# Manipulation of anisotropic Zhang-Rice exciton in NiPS$_3$ by magnetic field

Feilong Song[1,4], Yanpei Lv[1,2,4], Yu-Jia Sun[1,2], Simin Pang[1,2], Haonan Chang[1,2], Shan Guan[1], Jia-Min Lai[1,2], Xu-Jie Wang [3], Bang Wu[3], Chengyong Hu[3], Zhiliang Yuan [3] & Jun Zhang [1,2] ✉

The effect of external magnetic fields on the behavior of the Zhang-Rice exciton in NiPS$_3$, which captures the physics of spin-orbital entanglement in 2D XY-type antiferromagnets, remains unclear. This study presents systematic study of angle-resolved and polarization-resolved magneto-optical photo-luminescence spectra of NiPS$_3$ in the Voigt geometry. We observed highly anisotropic, non-linear Zeeman splitting and polarization rotation of the Zhang-Rice exciton, which depends on the direction and intensity of the magnetic field and can be attributed to the spin-orbital coupling and field-induced spin reorientation. Furthermore, above the critical magnetic field, we detected additional splitting of the exciton peaks, indicating the coexistence of various orientations of Néel vector. This study characterizes orbital change of Zhang-Rice exciton and field-induced spin-reorientation phase transitions in a 2D hexagonal XY-type antiferromagnet, and it further demonstrates the continuous manipulation of the spin and polarization of the Zhang-Rice exciton.

Two-dimensional (2D) magnetism, which is associated with strong intrinsic spin fluctuations, has long been the heart of fundamental questions in condensed matter physics[1,2]. The recent emergence of van der Waals (vdW) magnet provides a new platform to explore spin-correlated physics, such as strong correlation, quantum magnetism, topological spintronics as well as their opto-spintronic and quantum transduction applications[3–5]. Yet, the study of exciton coupled with the intrinsic magnetic order in 2D magnet which plays an important role in bridging the photon degree with other quasiparticles in condensed matter such as magnon, phonon, etc., remains a challenge due to the lack of suitable materials. Recently, spontaneous circularly polarized photoluminescence (PL) from the Frenkel excitons in ferromagnetic monolayer CrI$_3$ and CrBr$_3$[6,7], and linear polarization exciton strongly coupled with coherent magnons in CrSBr have been observed[5]. However, the polarization of the exciton is insensitive to the magnetic field direction, which means that little anisotropic information of magnetic order is reflected from the exciton[8].

NiPS$_3$, as a 2D vdW antiferromagnetic insulator, belongs to a class of transition metal phosphorus trichalcogenides (APX$_3$, A: Fe, Mn, Ni and X:S, Se). Recent reports have shown that the X$_1$ exciton in NiPS$_3$ exhibits strong excitonic anisotropy in the optical linear dichroism (LD) spectrum, pump-probe Kerr rotation experiment, PL spectrum, etc[9–13]. Different from the CrBr$_3$ and CrI$_3$[6,7], the X$_1$ exciton emission in NiPS$_3$ exhibits an ultra-narrow linewidth (about 350 μeV) and highly linear polarization[10,12,14]. In addition, the exciton tends to couple with other quasiparticles. For example, the LD spectra show that the exciton is coupled with the $A_{1g}$ phonon and forms the exciton-phonon bound state[10]. Moreover, the NiPS$_3$ thin flake in a photonic cavity shows that the exciton is strongly coupled with the cavity photon[15]. However, further applications of the exciton, such as the quantum manipulation and opto-spintronic devices, are hindered by several problems. Firstly, the origin of the X$_1$ exciton remains controversial, like the Zhang-Rice exciton (ZRE)[14], the Wannier-like exciton from the recombination of electron and hole[10,16], the spin-flip induced *d-d* emission in the Frenkel

[1]State Key Laboratory of Superlattices and Microstructures, Institute of Semiconductors, Chinese Academy of Sciences, Beijing, China. [2]Center of Materials Science and Optoelectronics Engineering, University of Chinese Academy of Sciences, Beijing, China. [3]Beijing Academy of Quantum Information Science, Beijing, China. [4]These authors contributed equally: Feilong Song, Yanpei Lv. ✉e-mail: zhangjwill@semi.ac.cn

exciton[12,17], the defect-related exciton[18], etc. Then, the relationships of the Néel vector and polarization of the exciton were identified variously[10,12,14]. Lastly, there is a lack of understanding and observation about energy shift or splitting of the $X_1$ exciton under the magnetic field, which is powerful in determining the characteristics of the exciton such as Zeeman splitting, diamagnetic shift, g-factor, etc[19,20]. It is essential for further understanding of spin-orbital coupling and magnetic transition such as spin-flop transition and application in opto-spintronics[3,21].

In this work, we conduct angle-resolved and polarization-resolved magneto-optical experiments of NiPS$_3$ in the Voigt geometry. With applying an in-plane magnetic field ($\mu_0\mathbf{H}$), we observe the Zeeman splitting of the linear polarized PL of the $X_1$ exciton. With rotating the sample in the **ab**-plane and changing the $\mu_0H$, we observe that the Zeeman splitting is highly anisotropic and non-linearly dependent on the $\mu_0H$. With the ZRE model, we attribute the anisotropy and non-linearity to magnetic field-induced spin reorientation transition (SRT) and read out the Néel vector in NiPS$_3$. With measuring the polarization of splitting PL peaks, we observe the identical anisotropic behavior of the polarization and confirm the spin-polarization coupling during the SRT process, which reflects the localism and orbital change of ZRE and provide an explanation for the short lifetime of ZRE. We accomplish the accurate determination of the spin orientation based on the polarization and splitting energy of the ZRE. Based on this spin read-out technique, we observe the coexistence of various orientations of Néel vector near the critical magnetic field approximately parallel with the **a**-axis. Our work demonstrates that the ZRE in NiPS$_3$ is an ideal platform for the study of the spin-orbital entanglement and the magnetic phase transition in the XY-type antiferromagnets. Besides, we also provide a method to continuously manipulate and detect the spin and the polarization of the ZRE, which enables the potential applications of antiferromagnets in quantum information and opto-spintronic devices.

## Results and discussion

Bulk NiPS$_3$ has a monoclinic structure with point group $C_{2h}$, whereas monolayer NiPS$_3$ has a trigonal structure with point group $D_{3d}$[22,23]. Figure 1a indicates the crystalline axis of the honeycomb lattice structure with three-fold rotational symmetry in monolayer NiPS$_3$. Below the Néel temperature ($T_N = 155$ K), NiPS$_3$ becomes an antiferromagnetic insulator[10,16], in which the Néel vector is confirmed almost aligned in-plane and along the **a**-axis[24]. The interlayer coupling in NiPS$_3$ is ferromagnetic, as shown in Fig. 1b. X-ray absorption studies suggest that NiPS$_3$ is a charge-transfer antiferromagnetic insulator that exhibits strong spin-charge correlation effects below the Néel temperature[25].

With the experimental setup as shown in Method and SI Fig. S1, we measured the temperature-dependent and magnetic field-dependent spectra. Figure 1c is the temperature-dependent reflection contrast spectra of the NiPS$_3$ thin flakes, in which there are two sharp peaks at 1.476 eV and 1.498 eV below the Néel temperature. The low-energy one (labeled as $X_1$) corresponds to the reported spin-correlated exciton. The high-energy one (labeled as $X_3$) was earlier attributed to the two-magnon sideband or Rydberg exciton of the $X_1$[14,16], while subsequent temperature-dependent experiments identified the $X_3$ as the phonon sideband of the $X_1$[12,23,26]. The temperature dependence of the reflection contrast spectra indicates that these two peaks are coupled to the long-range magnetic order as the two peaks get considerably weakened and disappear above 150 K when the temperature increases. And the temperature-dependent polarization-resolved PL spectra (as shown in SI Fig. S2) also indicate similar magnetic behavior with 2D XY-type systems[12,27,28]. Figure 1d shows the polarization-resolved PL spectra of NiPS$_3$ thin flakes at 4 K, from which we can easily find that the PL is almost linearly polarized. The sharp PL peak (labeled as $X_1$) is observed at 1.4758 eV with a side peak (labeled as $X_2$) at 1.4788 eV. As shown in SI Fig. S3, by varying the intensity of the laser and the sample position, the PL intensity of $X_1$ is linearly dependent on the laser power

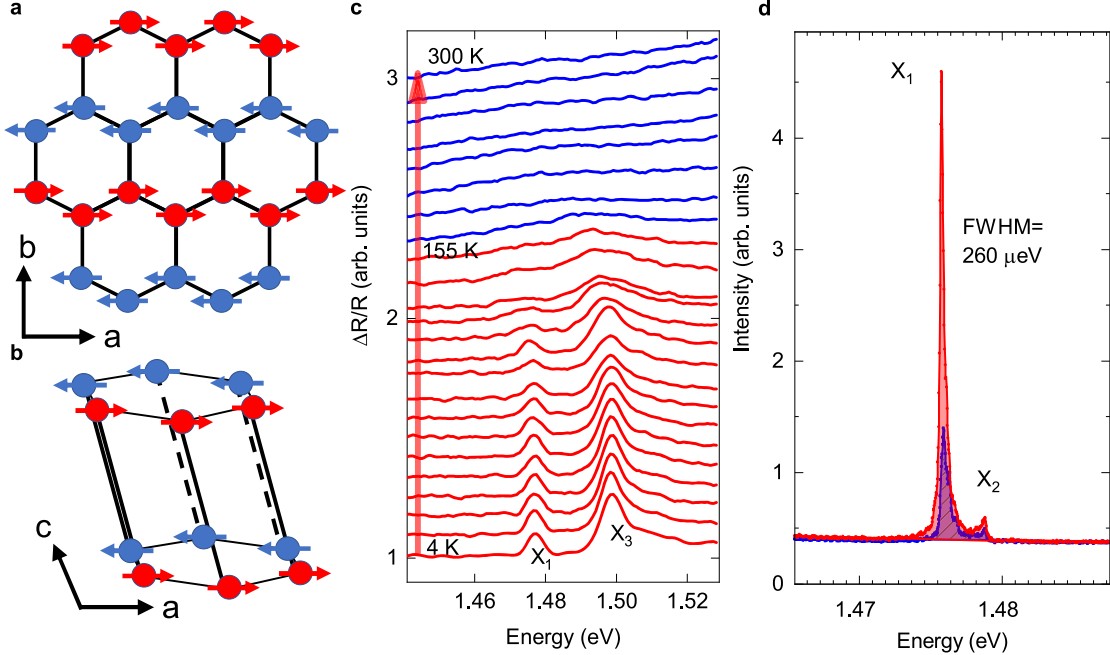

**Fig. 1 | Crystal structure and antiferromagnetic exciton state of NiPS$_3$.**
**a** Schematic of the spin orientation in a single layer of NiPS$_3$. **b** Monoclinic stacking with ferromagnetic interlayer coupling in a bulk crystal. **c** Temperature-dependent reflection contrast spectra. The red curve denotes the reflection contrast spectra below the Néel temperature, while the blue curve is above Néel temperature. These peaks in the reflection contrast spectra are assigned to the $X_1$ and $X_3$ exciton. **d** Polarized PL spectra collected from the parallel and perpendicular directions to PL polarization, the linewidth of the PL spectra is about 260 μeV. The red line is the PL spectrum collected parallel to the PL polarization, while the blue line is perpendicular to the PL polarization. In the spectra, there is a main peak position at 1.476 eV (labeled as $X_1$) and a shoulder peak position at 1.479 eV (labeled as $X_2$).

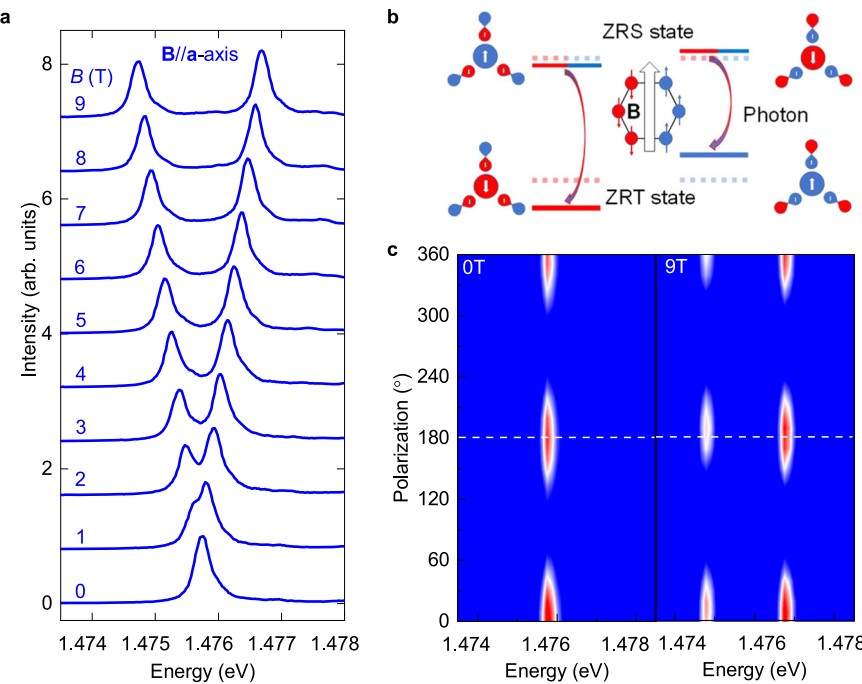

**Fig. 2 | Zeeman splitting of the excitons in NiPS₃ thin flakes. a** Magnetic field-dependent PL spectra and Zeeman splitting of ZRS exciton in NiPS₃ thin flakes. **b** A schematic view of the Zeeman splitting of the ZRE under the magnetic field parallel to the **a**-axis. **c** Contour plot spectra of the polarized PL spectra with a magnetic field at 0 T and 9 T (left panel and right panel respectively). The direction of the magnetic field is parallel to **a**-axis.

and the profile of the $X_1$'s PL is insensitive to the sample position. Hence, the defect-related origin of $X_1$ could be excluded. The $X_1$ exciton shows an ultra-narrow linewidth in our experiments, about 260 μeV (Fig. 1d), similar to those of other groups[10,12,14], which indicates the good crystallization of our sample and the suitability for studying the exciton properties with the magneto-optical experiment[29,30].

Up to now, number of phenomena of this sharp exciton emission have been fully researched with magneto-optical experiments[12,31,32], except for the Zeeman splitting. The magnetic field will induce the Zeeman splitting of the excitons in traditional semiconductors when its component along the spin direction of the excitons is nonzero[30,33,34]. Hence, we apply a magnetic field in the **ab**-plane of NiPS₃ to detect the magneto-optical properties of $X_1$. We observe that the $X_1$ exciton splits into two peaks when $\mu_0\mathbf{H}$ is parallel to the **a**-axis, as shown in Fig. 2a. The splitting energy is linearly dependent on the $\mu_0 H$ with a slope of 0.22 meV/T, as shown in SI Figure S4. When the magnetic field is increased from 0 T to 9 T, the PL peaks of ZRE keep linearly polarized and the polarization keeps parallel with the external field, as shown in the Fig. 2c. This polarization behavior indicates that the PL of ZRE is a π light which corresponds to the unchanged total angular momentum from the excited state to the ground state[35,36]. The Zeeman splitting of the $X_1$ can be explained by a model as shown in Fig. 2b. The ground state (Zhang-Rice triplet, ZRT) has a high-spin state of $S = 1$ while the spin of the Zhang-Rice singlet (ZRS) is zero[14]. The π light of ZRE means the contribution of spin-orbital coupling and the change in orbitals from ZRS to ZRE. As a result, the difference of gyromagnetic ratio ($\Delta_g$) between the ZRT and ZRE leads the energy of ZRE in different spin chains to shift in an opposite direction under the magnetic field along the **a**-axis, as discussed in Methods, SI Note 4, and SI Fig. S6. The orbital change means that the spin-correlated exciton does not originate from $d$-$d$ transition and the exciton could obey optical selection rules, which could provide potential explanation for the short lifetime of ZRE[10,12], as discussed in the SI Note 5. The **a**-axis could be determined with an angle-resolved measurement as discussed in Methods and SI Note 1.

Since the ZRE is coupled with magnetic order and its PL spectrum exhibits an ultra-narrow linewidth, the Zeeman splitting could accurately indicate the information of the Néel vector. Similar protocols to detect and manipulate Néel vector have been accomplished with the Zeeman effect of magnons in bulk crystals such as MnF₂ and Cr₂O₃[21,37]. As shown in Fig. 3a, we systematically measured the magnetic field-dependent PL spectra of NiPS₃ thin flakes. The $\theta$ denotes the angle between $\mu_0\mathbf{H}$ and the **a**-axis. In the first panel ($\theta = 4°$) of the Fig. 3a, a typical Zeeman splitting phenomenon with the magnetic field below 10 T can be observed, where the peaks of the exciton linearly split with $\mu_0 H$. When the $\theta$ increases from 4° to 86°, the Zeeman splitting shows highly anisotropic and nonlinear dependences on the $\mu_0 H$, which could be illustrated by the ZRE model as discussed above. The anti-paralleled magnetic moments of ZRE have different Zeeman energies $\pm\Delta_g \mu_B \mu_0 H \cos\varphi$, where $\varphi$ represents the angle between the $\mu_0\mathbf{H}$ and the Néel vector. The nonlinearity and anisotropy of Zeeman splitting indicate that $\varphi$ depends on $\mu_0 H$ and $\theta$. In other word, a direct conclusion from the spectra based on the ZRE model is that the Néel vector will rotate to a stable orientation under the in-plane magnetic field. The similar spin-rotating phenomenon, which is called as the magnetic field-induced spin reorientation transition (SRT), have been reported by Néel and Yosida in the research of the tetragonal MnF₂[38,39].

To further research the dependence of Zeeman splitting on the $\mu_0\mathbf{H}$, we extracted the Zeeman splitting from the spectra in Fig. 3a and fitted them with a formula $2\Delta_g \mu_B \mu_0 H \cos\left(\theta - \frac{1}{2}\arctan\left(\frac{\sin 2\theta}{\cos 2\theta - \frac{H_c^2}{H^2}}\right)\right)$, as shown in Fig. 3b, where $\mu_0 H_c$ is selected as 10.75 T, consistent with the reported experimental result[12]. We should note here that the fit formula, which could be directly obtained from our data as shown in Methods, is just the same as Néel's formula in the discussion of MnF₂[39]. In Fig. 3c, the $\varphi - \theta$ represents the rotation angle of the Néel vector, which is extracted from the Fig. 3b and fitted with $\varphi = \theta - \frac{1}{2}\arctan\left(\frac{\sin 2\theta}{\cos 2\theta - \frac{H_c^2}{H^2}}\right)$. With Monte Carlo simulations for a 2D

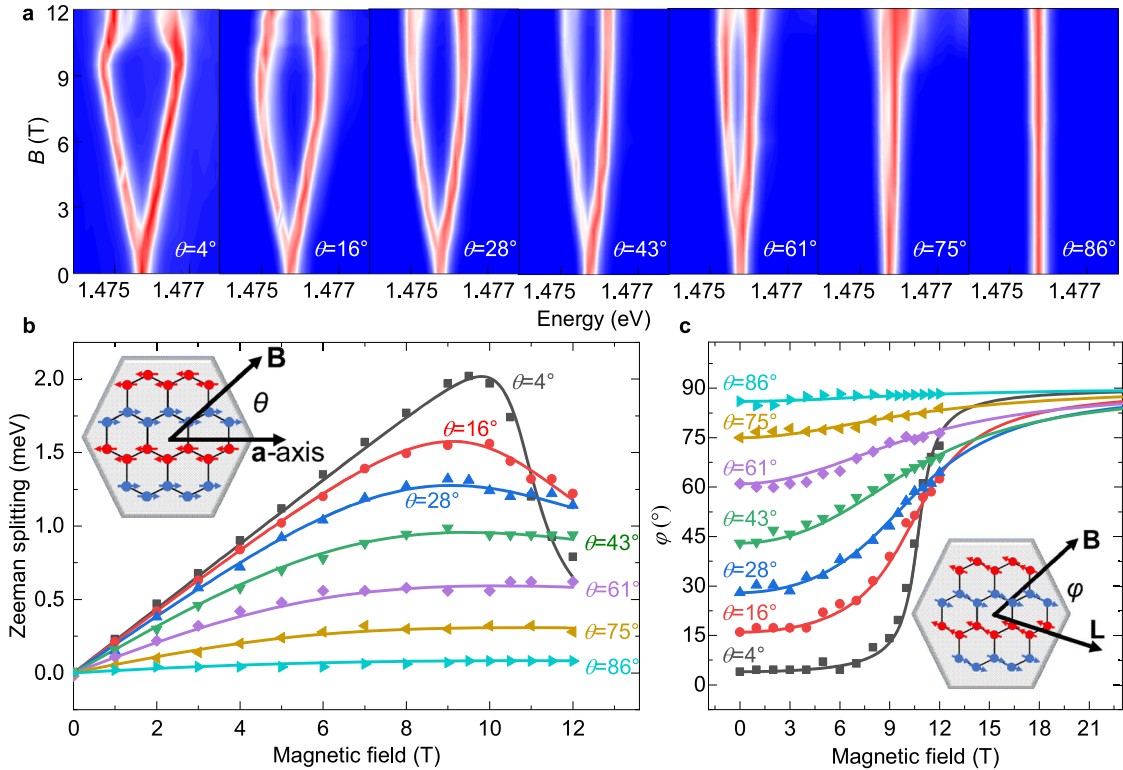

**Fig. 3 | Angle-dependent Zeeman splitting of ZRE. a** Contour plot of the magnetic field-dependent PL spectra of ZRE under different $\theta$. $\theta$ is the angle between the $a$-axis and the magnetic field. **b** The Zeeman splitting energy as a function of the angle $\theta$ and magnetic field. The dots are the experimental data, while the lines are the corresponding fitted results of the same-colored data. **c** Magnetic field-dependent Néel vector rotation. The dots are calculated from the experimental data, while the lines are the corresponding fitted results. The inset cartoon schematically illustrates the angle $\varphi$ between magnetic field and the Néel vector **L**.

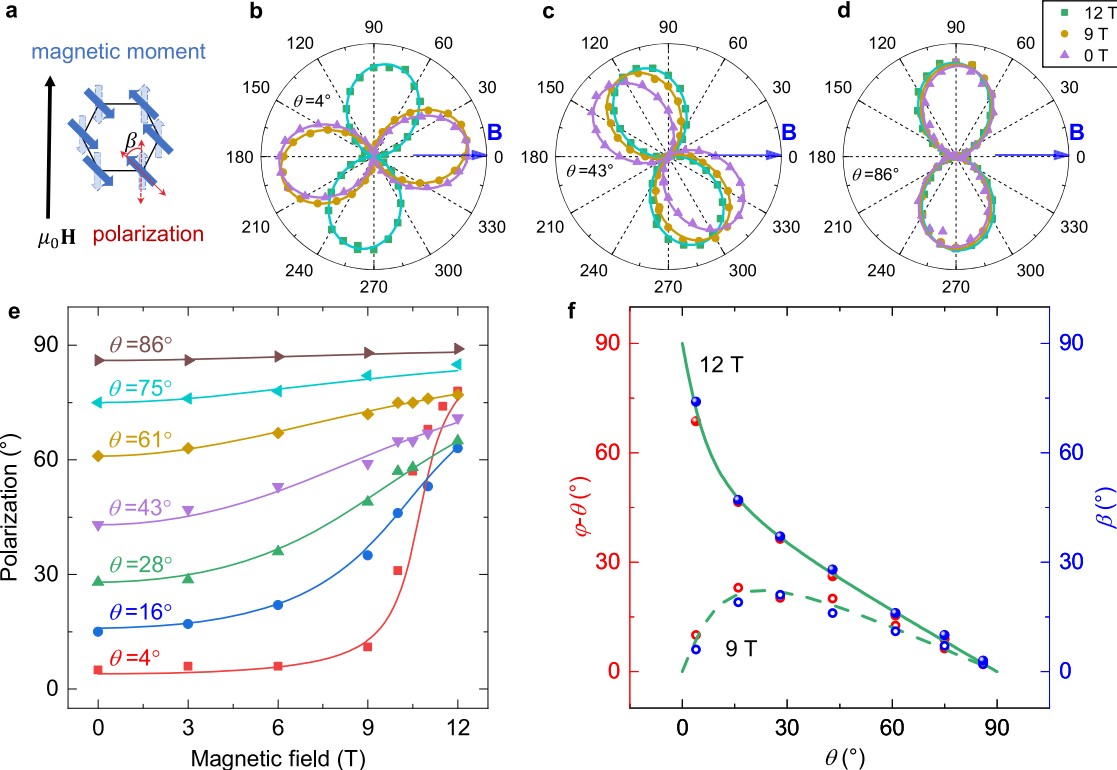

**Fig. 4 | Anisotropic rotation of the polarization of ZRE under the magnetic field from 0 T to 12 T. a** The $\beta$ is defined as rotation angle of polarization in ZRE with and without magnetic field. **b–d** The rotation of the polarization of ZRE at a magnetic field of 0 T, 9 T and 12 T under $\theta$ is about 0°, 45°, and 90°. **e** The scattering points are measured polarizations and the curves are the same as curves in Fig. 3c. **f** Anisotropic rotation of the polarization of ZRE and the Néel vector when the magnetic field is 9 T and 12 T. The red and blue small balls represent the data of $\varphi - \theta$ and $\beta$ respectively. The dash and solid line are calculated from Néel's formula.

hexagonal XY-type antiferromagnet (as shown in Methods and SI Fig. S5), the simulated Zeeman splitting and $\varphi$ are the same as the curves calculated with Néel's formula in Fig. 3 b and c, reflecting the anisotropy akin to a tetragonal lattice under a magnetic field for the zig-zag antiferromagnets. Due to the narrow linewidth of NiPS₃, we obtain small errors in the extracted energies and the angle $\varphi$, which indicate the potential application of ZRE for accurate spin readout. With the $\mu_0 H$ ranging from 0 T to 12 T, we realize the manipulation of the Néel vector to almost any direction in the **ab**-plane of NiPS₃, as shown in Fig. 3c.

Furthermore, we varied the $\mu_0 H$ and $\theta$ to measure the polarization of ZRE to study the anisotropic behavior of it. To research the rotation of polarization, we defined the polarization-rotation angle $\beta$, as shown in Fig. 4a. Figure 4b shows that the polarization of ZRE is unchanged below the critical magnetic field and rotates about 90° above the critical magnetic field, when $\theta$ is 0°. Figure 4c shows that the polarization continuously rotates when $\theta$ is 43° and Fig. 4d shows that nearly no polarization rotates when $\theta$ is 86°, when the magnetic field increases from 0 T to 12 T. In Fig. 4e, we extract the polarization angle for different $\theta$ under the $\mu_0 H$, increasing from 0 T to 12 T. We find that the polarization angle can be described by the same formula that characterizes the reorientation of the Néel vector under the same $\mu_0 \mathbf{H}$, as shown in Fig. 4e, indicating the spin-polarization coupling of ZRE during the SRT process. Figure 4f is the comparison of the extracted $\beta$ and the $(\varphi - \theta)$ at 9 T and 12 T, which confirms that the rotation of the Néel vector and polarization is identical under the magnetic field below and above the $\mu_0 H_c$. Interestingly, the distorted sine function in Fig. 4f at 9 T is similar to that in magnetic transport experiments[40].

To research the behaviors of the SRT at the critical magnetic field, we conducted magnetic field-dependent polarization measurements for ZRE when $\theta = 1°$, as shown in Fig. 5a. When the $\mu_0 H$ increases from 0 T to 9.5 T, the exciton peaks split into two branches without

polarization rotation. However, when the magnetic field increases to 10 T, the energy and polarization of the exciton peaks that existed below 9.5 T remain almost unchanged, while two new peaks appear with smaller splitting energy and polarizations nearly perpendicular to the original peaks. For clarity, the new emerging peaks are indicated by green arrows (dots and curves) and pre-existing peaks are indicated by red arrows (dots and curves), as shown in Fig. 5a (Fig. 5 b–e). As shown in the Fig. 5a, d, and e, when increasing the magnetic field from 10 T to 12 T, the intensity of the pre-existing peaks become weaker, while the intensity of the new emerging peaks becomes stronger, indicating the transformation of different spin configurations. Besides, the polarization and energy of the pre-existing peaks gradually get closer towards the new emerging peaks, while the polarization and energy of the new emerging peaks remain essentially unchanged, indicating that the spin-polarization coupling is still maintained. The multiple peaks reflect the change of magnetic order in the transition region near the critical field. This phenomenon could originate from inhomogeneity-induced various spin-flop critical fields[41] or the transition to an intermediate phase[42]. If the phenomenon originates from inhomogeneity, our findings can help to further investigate the origin of inhomogeneity and thus domain engineering[43] in antiferromagnetic materials. If the phenomenon originates from the intermediate phase, the neutron scattering could help to further investigation[42] and the magnetic field dependence of the peak positions and polarization in Fig. 5 could contribute to the study of magnetic phase transitions in XXZ-type or XY-type antiferromagnets (AFM)[23]. Further determination of the origin needs more experimental evidence and is beyond the scope of our discussion. Our results show that the PL polarization and energy of ZRE is a non-destructive and convenient method to detect the critical behavior of NiPS₃.

From a systematically angle-resolved and polarization-resolved magneto-optical PL spectra measurement, we observed non-linear and

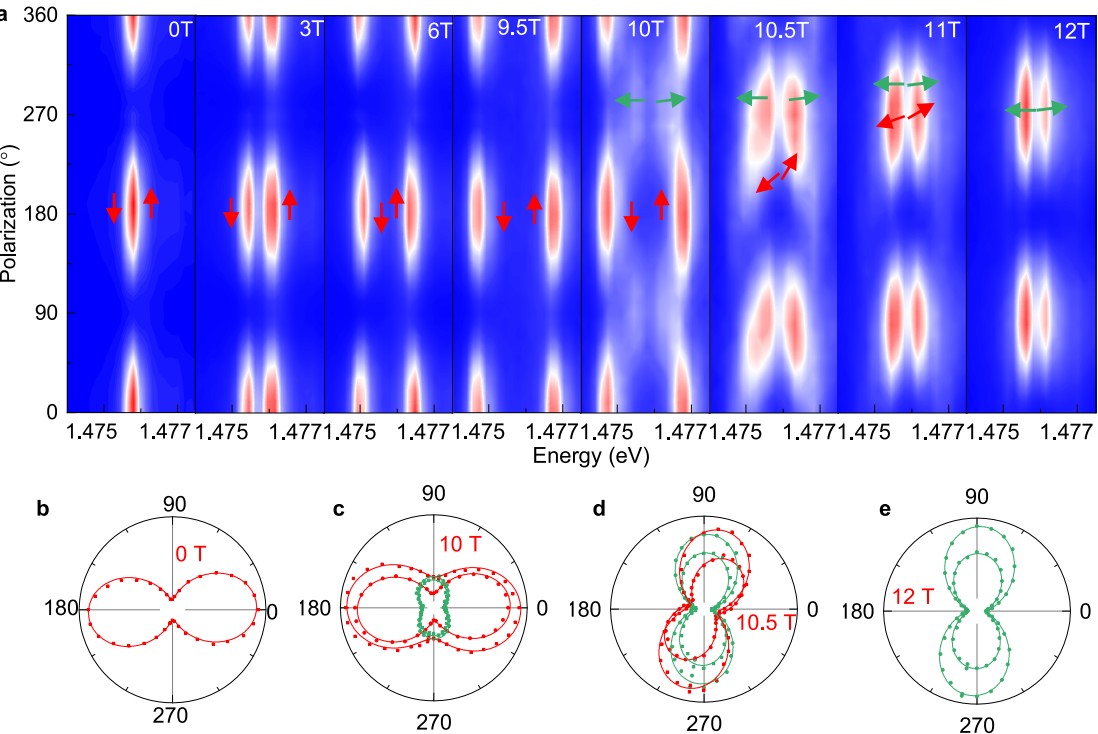

**Fig. 5 | Field-induced evolution of multiple spin configurations of ZRE.**
**a** Polarization-resolved PL spectra of ZRE when $\theta$ is approximately 1°. The four arrows represent the different spin orientations extracted from the four exciton peaks. Red arrows represent the pre-existing peaks, while green arrows represent the new emerging peaks. **b–e** Polarizations extracted from the four peaks. Dots are extracted polarization from experimental data, while the lines are fitting curves. Red dots represent the pre-existing peaks, while green dots represent the new emerging peaks.

anisotropic Zeeman splitting originated from magnetic field-induced SRT. Mapping the energy splitting to the direction of the Néel vector, we confirmed that the anisotropic linearly polarized PL emission of ZRE in NiPS$_3$ is coupled with Néel vector when varying the in-plane magnetic field, which indicates the spin-orbital coupling and orbital change from ZRS to ZRT. With the advantage of the ultra-narrow linewidth, we found that the orientation of the Néel vector could be continuously read out and manipulated by an external in-plane magnetic field, which opens a path for detecting and manipulating the spin-orbit-entangled exciton state in a 2D magnetic material and promotes the development of opto-spintronic devices and magnetic quantum information technology. With this spin-readout technique, we probe the coexistence of various orientations of the Néel vector, which could be attributed to the inhomogeneity or the transition to an intermediate phase. Our findings provide new insights for the localism, orbital change, and short lifetime of ZRE, as well as AFM phase transition in NiPS$_3$. We thus foresee that NiPS$_3$ is a promising 2D platform for exploring magnetic phase transition and strongly correlated systems with the optical methods. After this article was submitted, we became aware of another paper[44] reporting related to the Zeeman splitting of NiPS$_3$.

## Methods

### Sample fabrication and PL measurement

The sample in our experiment was mechanically exfoliated from the high-quality NiPS$_3$ bulk single crystals grown by chemical vapor transport method. It was then transferred to a marked SiO$_2$/Si substrate. A He-Ne laser (633 nm) was used to excite the NiPS$_3$ sample, which was placed in a closed cycle cryostat (attodry 2100). The PL spectra of the exciton in NiPS$_3$ were polarization-resolved and collected by a spectrometer with a liquid nitrogen-cooled CCD. A magnetic field up to 12 T was applied in Voigt geometry. A low-temperature objective was used to focus the excitation laser onto the sample and collect the PL signal. An electronically controlled rotary stage enables us to rotate the sample in a low-temperature and high-magnetic-field environment. The magneto-optical experiments were performed in Voigt geometry. The sample was positioned vertically inside the magnetic cell with the surface parallel to the applied magnetic field for the Voigt geometry. A mirror was set between the objective and sample with an angle of 45° to change the optical path by 90°. For both the polarization-dependent and magnetic-field-dependent experiments, the PL signal was filtered by an analyzer and then collected and measured by a spectrometer with a CCD. Two samples were measured in our experiments, as shown in SI Note 1.

### Experimental details

The images of the sample inside the magnetic cell were captured with a camera, as shown in the SI Fig. S7. The polarization-resolved spectra, when $\theta$ ranges from -30° to 90° (SI Fig. S8), show that the Zeeman splitting when $\theta=0°$ is the largest. Combining the images, the Zeeman splitting, and the polarization, we could determine $\boldsymbol{a}$-axis as discussed in the SI Note 1. By flipping the magnetic field direction, the Faraday rotation effect could be excluded in the research of polarization rotation, as shown in the SI Fig. S9. Before and after flipping the $\mu_0\mathbf{H}$, the behaviors of SRT were consistent when the $\mu_0H$ is higher than the $\mu_0H_c$, as shown in Fig. S10.

### ZRE model and data analysis

Although the total angular momentum of the ground state and excited state is the same ($\mathbf{J_{ZRT}}=\mathbf{J_{ZRS}}\equiv\mathbf{J_0}$), there is a distinction in the quantum numbers of spin and orbital angular momentum between the ground and excited states, as discussed in the SI Note 4. Consequently, the effective gyromagnetic ratios of the ground state and excited state are

typically different, that is, $g_{ZRT}\neq g_{ZRS}$. In antiferromagnetic materials, the localized magnetic moment is represented by $\mu_B g\mathbf{J_0}$ or $-\mu_B g\mathbf{J_0}$, where $\mu_B$ denotes Bohr magneton. Under the external magnetic field $\mathbf{B}$, the exciton ground state energy is given by $E_{ZRT}(\mathbf{B})=E_{ZRT}(0)\pm\mu_B g_{ZRT}\boldsymbol{J_{ZRT}}\cdot\mathbf{B}$, while the corresponding exciton excited state energy is $E_{ZRS}(\mathbf{B})=E_{ZRS}(0)\pm\mu_B g_{ZRS}\mathbf{J_{ZRS}}\cdot\mathbf{B}$. Therefore, the peak positions of the ZRE is $E_{ZRE}(\mathbf{B})=E_{ZRE}(0)\pm\mu_B(g_{ZRT}-g_{ZRS})\mathbf{J_0}\cdot\mathbf{B}=E_{ZRE}(0)\pm\mu_B\Delta_g\mathbf{J_0}\cdot\mathbf{B}$. In the external magnetic field, the "spin reorientation" means that the localized magnetic moment ($\pm\mu_B g\mathbf{J_0}$) rotates in the plane, leading to anisotropic and nonlinear Zeeman effects, as stated in the original version. The peak positions in Fig. 3b are extracted from PL spectra in Fig. 3a with peaks fitted by the Lorentzian function. The angles ($\varphi-\theta$) are extracted from the splitting energy in Fig. 3b with formula $\varphi-\theta=\arccos\left(\frac{\Delta E(\theta,\mu_0H)}{2g\mu_B}\right)$, whose errors include one part propagated from errors of peak positions and another part resulted from angle measurement in experiments. The fit curves in Fig. 3b and Fig. 3c are both plotted based on the following steps. Firstly, as shown in SI Fig. S11a or c, we plot the dependence of $\varphi-\theta$ on $\theta$ under different $\mu_0H$. The dependence is well fitted with formula $\varphi-\theta=-\frac{1}{2}\arctan(\frac{\sin 2\theta}{\cos 2\theta-A})$ and the fitting parameter $A$ are various in different $\mu_0H$. Then, as shown in SI Fig. S11b or d, we plot the dependence of $A$ on $\mu_0H$, which is fitted with $A=\frac{H_c^2}{H^2}$ and $H_c=10.75$ T. As a result, we could plot the dependence of $\varphi-\theta$ on $\mu_0H$ for any $\theta$. Finally, combined with the Zeeman splitting energy $\pm g\mu_B\mu_0H\cos\varphi$, we could plot the dependence of splitting energy on $\mu_0H$ for any $\theta$. The formula $\varphi-\theta=\frac{1}{2}\arctan\left(\frac{\sin 2\theta}{\cos 2\theta-\frac{H^2}{H_c^2}}\right)-\theta$, obtained from our data here, is different from Néel's formula, while it is consistent in physics for the $C_2$ symmetry of NiPS$_3$. More discussions could refer to SI Note 2.

### Monte Carlo simulations

We calculated the zig-zag antiferromagnetic spin evolution at 1.6 K by Monte Carlo simulations. We treated the spins classically and included the short-range exchange interactions up to third neighbors with ignoring the anisotropic exchange interaction. At each magnetic field, total steps are set as 30000. For all the numerical data from Monte Carlo simulations, the numerical uncertainties are smaller than or comparable to the size of symbols. More details are indicated in SI Note 3.

## Data availability

The authors declare that data generated in this study are provided in the paper and the Supplementary Information file. Further datasets are available from the corresponding author upon request. Source data are provided in this paper.

## Code availability

The authors declare that code supporting the findings of this study is available within the paper and the Supplementary Information file. Further code is available from the corresponding author upon request.

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

## Acknowledgements

We are grateful for high-field measurements supported by Xiamen University (X. Wang) and helpful discussion about the theoretical calculations with University of Science and Technology of China (J. Zhao). Authors acknowledge the funding support from Chinese Academy of Sciences - the Scientific and Technological Research council of TÜRKİYE Joint Research Projects (172111KYSB20210004), the CAS Interdisciplinary Innovation Team, National Natural Science Foundation of China (12074371), Research Equipment Development Project of Chinese Academy of Sciences (YJKYYQ20210001).

## Author contributions

J.Z. supervised the project; J.Z. and F.S. conceived the ideas; F.S. and Y.L. prepared the samples and carried out the experiments, assisted by H.C., Y.S., X.W. and B.W. F.S., Y.L. and J.Z. analyzed the data. Y.L. fitted the anisotropic Zeeman splitting data. S.G. and Y.L. conducted the Monte Carlo simulations. F.S., Y.L. and J.Z. wrote the paper with input from all authors. All authors discussed the results and commented on the manuscript.

## Competing interests

The authors declare that they have no competing interests.
