## [Peer Review File · Nature Communications]

Reviewers' Comments:

Reviewer #2:

Remarks to the Author:

In this work, Song and coworkers report on the behavior of the so-called Zhang-Rice exciton in NiPS3 under magnetic field. They observe the splitting of the exciton peak when the magnetic field is along the a-axis. They further study the effect on the exciton peak when the magnetic field is rotated in the plane of the sample and propose a 'spin-dipole separation'. The rotation of the polarization of the exciton under an in-plane magnetic field has been reported in Ref. 4, and so the novelty of this work is limited. The measurement of the Zeeman splitting with the in-plane magnetic field is new, but I am not sure if it is enough for the work to be published in Nature Communications. I leave the decision onto the editors.

There are a few technical comments the authors should consider.

1. The English should be improved. Some examples of grammatical errors are: line 92 "corresponds to the report spin-correlated..."; line 146 'from that the anti-paralleled spins...?'
- 2, How was the a-axis determined? Since the direction of the a-axis is the most important parameter of this work, it should be carefully determined before any analysis is done. Was it measured by x-ray or electron diffraction on the same sample?
3. In Figure 3, the Néel vectors are indicated. However, this is based on a rather simple model calculation. Since the direction of the Néel vector and the exciton polarization is the main topic of this work, the Néel vector direction should be determined from a separate measurement such as neutron scattering.
4. In Figure 3, the experimental data agrees with the calculated curves only in the linear range. For example, the Zeeman splitting in Fig. 3b does not show the decrease with the magnetic field in the high-field range.
5. The argument for the 'spin-dipole separation' is not convincing. The only experimental observation is that the polarization of the exciton peak rotates with the in-plane magnetic field even when there is no Zeeman splitting. What does it mean that the spin and the dipole are separated? Does it mean that the exciton is separated from the magnetic ordering? If so, the temperature dependence of the exciton peak under in-plane magnetic field should be measured for different rotation angles.

Reviewer #3:

Remarks to the Author:

The Zhang-Rice exciton (ZRE) in NiPS₃ is an emerging observation, but not fully understood yet. This work performed the magneto-optical measurements on NiPS₃ in a Vioget geometry, and observed the splitting of the ZRE under magnetic field ranging from 0 to 9 T. They further reported that the splitting is anisotropic depending on the direction of the applied B field. The mechanism behind the splittings is well discussed. But I also find some confusing points and some unconvincing conclusions in this work to be addressed before publishing on Nature Comm.

In the introduction, the authors said “there is no report about energy shift or splitting of the ZRE under magnetic field...” which is not true. This paper on Arxiv reported the energy splitting of the exciton (<https://arxiv.org/pdf/2306.07660.pdf>), although there is still lack of understanding.

In the introduction, the authors said “...the PL linewidth of the exciton is broad (roughly 150 meV), which is not desirable for magneto-optical physics”. I am not sure if I agree with this. There are lots of things in magneto-optical physics to be studied. I think there could be still interesting magneto-optical physics in broad excitons.

The authors also mentioned that there are many controversies about the exciton properties in the introduction. For example, there are reports on the assignment of the exciton at 1.47 eV to both from the material itself and from the defects in the materials. It seems that the authors agree that this is an unanswered question, but then on page 3 (Figure 1d), the authors use the narrow bandwidth of the exciton to support the good crystallinity of the sample. The authors should at least give their evidence to support that the exciton is not from defects, so that they can make this conclusion.

In addition to the X1 peak, the authors also observed another sharp peak at 1.498 eV (X3). They cite reference 2 and 15 to say they are the two-magnon sideband. But there is another paper (Nature Materials volume 20, pages 964–970 (2021)) attributed it to a different origin. It would be better that the authors provide a more accurate picture to the readers on what the current understanding is.

The labels in the schematics in Figure 2 are not clear. For example, it is not indicated what is the red and blue balls, and what the arrows mean. The authors should be clear about the schematic presentation.

The authors claim that they observed spin-dipole separation (Figure 4), which is not a convincing conclusion. They got this conclusion based on that there is no exciton

splitting under B field with $\theta=90^\circ$, but the polarization of the exciton rotates with B field along $\theta=90^\circ$ direction. First, the reason they do not see the splitting could be because the spectra resolution is too low. Second, there could be a small error in determining the angle to be exact 90° . If we consider these two reasons, it will be hard to say that the spin and dipole are separate. In addition, the authors simply reported on the experimental observation (as I summarized briefly above) without in depth understanding on why spin-dipole separation is reasonable in NiPS₃. In fact, there are already quite a few papers which are also cited by this paper, including some other results presented in this work, showed that the spin the dipole are strongly coupled in NiPS₃. Thus, I am not convinced about the spin-dipole separation conclusion.

In the method section, when describing the spin-dipole separation, it is not easy to follow as there are some unexplained terms in the math and missing references. For example, in "...only the C66 and C13 term survive..." what are C66 and C13? The authors should check the math in this work careful and make it easy to follow.

How did the authors determine the a-axis of the crystal? One important conclusion the authors made is that the linear polarization of the exciton is along the a-axis of the crystal. But it is not clear how the authors determine the a-axis of the crystal.

The authors said that their results show we can use the exciton splitting for accurate spin read out. I understand the splitting is related to the angle between the Neel vector and the B field. But it would be better to provide another evidence from a difference angle to show that the Neel vector changed under B field, for example, a theoretical calculation or neutron scattering measurements (the neutron scattering might be difficult to do on small samples). Also, as we know there should be a magnetic phase transition at $B>10$ T, it would be better to show the data across the phase transition, so it would be more convincing.

Reviewer #2 (Remarks to the Author):

In this work, Song and coworkers report on the behavior of the so-called Zhang-Rice exciton in NiPS₃ under magnetic field. They observe the splitting of the exciton peak when the magnetic field is along the *a*-axis. They further study the effect on the exciton peak when the magnetic field is rotated in the plane of the sample and propose a ‘spin-dipole separation’. The rotation of the polarization of the exciton under an in-plane magnetic field has been reported in Ref. 4, and so the novelty of this work is limited. The measurement of the Zeeman splitting with the in-plane magnetic field is new, but I am not sure if it is enough for the work to be published in Nature Communications. I leave the decision onto the editors. There are a few technical comments the authors should consider.

Reply: We appreciate the thorough review and insightful suggestions provided by the reviewer, which have facilitated further improvements to our manuscript.

As the reviewer said, ref. 4 has reported polarization rotation under an in-plane field. In our study, we present three primary innovations in the angular dependence of the exciton’s polarization under an in-plane magnetic field:

1. Our observation of a novel angular dependence of photoluminescence (PL) polarization, as depicted in Figure 4 f, introduces a new magnetic orientation angle-dependent feature that follows a distorted sine function. This distinct non-trivial angular dependence, consistent with behaviors in anisotropic antiferromagnets, has garnered interest for various magnetic transport experiments, capturing the physics of field-driven rotational symmetry breaking of quasi-particles and offering potential spintronic applications¹⁻³. With a period of $\pi/2$ and a steeper slope near zero degrees, our data reveals the uniaxial anisotropy and potential for rapid polarization manipulation of the Zhang-Rice exciton (ZRE).

2. We have confirmed the spin-polarization coupling of the Zhang-Rice exciton at various magnetic field directions, spanning from 0° to 90°. The intricate angular dependences necessitate the validation of "spin-dipole coupling" across multiple angles⁴.

3. Our method of conducting angular measurements from 0° to 90° utilizing a low-temperature rotator at the same sample is new and useful. Given that temperature and sample location variations might impact the polarization orientation of NiPS₃, conducting measurements within the same sample at a low-temperature environment enhances reliability. Unlike other reported approaches, which involved simultaneously measuring two samples with different orientations or reloading the sample after changing the sample angle, our method utilized an in-situ rotator at low temperature and magnetic field, enabling measurements at multiple angles with the same sample location and improving the accuracy of the rotation angle.

Besides, as the reviewer mentioned, our original data on Zeeman splitting is new. Our revised version added more data with magnetic field up to 12 T and reported additional new phenomena. Our measurements of Zeeman splitting hold significance in the context of two-dimensional magnetic phase transitions, as well as in the manipulation and detection of spins in antiferromagnets.

1. Our Zeeman splitting data provides insights into the anisotropic and critical

behaviors of the Zhang-Rice exciton in the 2D antiferromagnet NiPS₃. The anisotropic, nonlinear Zeeman splitting, arising from field-induced spin reorientation reveals the strong coupling between magnetic order and the ZRE. Furthermore, the linear magnetization and uniaxial anisotropy, as indicated by the field-dependent exciton energy in Figure 3 **b**, reflect the anisotropic behavior of zig-zag antiferromagnet NiPS₃, similar to that in tetragonal-lattice antiferromagnets⁵. The observed further splitting at high fields in Figure 3 **a** and Figure 5 **a** reveals two different critical fields and the coexistence of various spin configurations, elucidating the outcome of the competition between different anisotropies. These observed novel phenomena of multi-state coexistence provide a deeper understanding of the correlated physics of the ZRE.

2. The Zeeman splitting results are significant for the detection and manipulation of spins in antiferromagnets. The angular dependence of the exciton energy splitting enables the detection of spin configurations and energy of the magnetic exciton⁶. The pronounced angular dependence of the spin-flop critical field establishes magnetic manipulation of anisotropy in antiferromagnets, reflecting potential anisotropic magnetic transport effects such as anisotropic magnetoresistance. Moreover, the coexistence of multiple spin configurations provides new insight for spintronic applications, such as multi-state magnetic storage^{2,7}.

In summary, through angle-resolved magneto-optical experiments, we have observed a series of novel phenomena in the PL spectrum and polarization of ZRE for the first time, which depend on the magnetic field vector. These phenomena include non-linear and anisotropic Zeeman splitting, polarization rotations, further splitting at high fields, and multi-angle consistency of polarization and Zeeman splitting of ZREs. We attribute these novel phenomena to the magnetic field-induced XY-type antiferromagnetic phase transition and the coupling of magnetic order, spin, and dipole of ZREs. Our study enables precise monitoring of the evolution of spin configurations in antiferromagnets and demonstrates continuous control of ZRE spins. These findings offer implications for magnetic phase transitions in two-dimensional antiferromagnets, the physics of correlated excitons, and the field of antiferromagnetic opto-spintronics.

Based on the valuable suggestions from the reviewer, we have conducted additional experiments and analyses, including:

- 1) Zeeman splitting at high magnetic fields up to 12 T,
- 2) Polarization-dependent PL spectra at high magnetic fields up to 12 T,
- 3) Monte Carlo simulation of the spin orientations of NiPS₃ under a magnetic field ranging from 0 T to 12 T.

Comments 1: The English should be improved. Some examples of grammatical errors are: line 92 “corresponds to the report spin-correlated...”; line 146 ‘from that the anti-parallelled spins...’.

Reply: We apologize for the language problems in the original manuscript. We have thoroughly edited the revised manuscript and appreciate the reviewer for pointing out these English grammar errors in our manuscript.

Comments 2: How was the *a*-axis determined? Since the direction of the *a*-axis is the most important parameter of this work, it should be carefully determined before any analysis is done. Was it measured by X-ray or electron diffraction on the same sample?

Reply: We thank the reviewer for raising this question. We would like to provide the following discussions regarding the determination of the direction the *a*-axis.

Below the Néel temperature, the direction of the Néel vector of NiPS₃ is along the *a*-axis of the crystal, which has been confirmed by neutron scattering and X-ray diffraction experiments as shown in the reference⁸. This result has been widely accepted by many works such as references^{4,9-11}. Due to the small size of our sample, conducting X-ray or electron diffraction measurements on the same sample presents challenges. Therefore, we determine the *a*-axis by ascertaining the direction of the Néel vector at 0 T with the application of an in-plane magnetic field. This is achieved through ZRE Zeeman splitting, ZRE polarization, and linear dichroism (LD) spectra for convenience.

The optical setup of our angle-resolved and polarization-resolved measurements is depicted in the Figure R1 **a**. We rotated the sample using an in-situ rotator while applying an in-plane magnetic field to the sample. In the angle-resolved PL experiments conducted from 0 T to 10 T, we observed that the Zeeman splitting of ZRE reached its maximum and exhibited high linearity in a specific orientation of the sample, as shown in Figure R1 **b**. Conversely, in the sample orientation perpendicular to this, the Zeeman splitting was almost negligible, as shown in the Figure R1 **c**. As the Zeeman splitting is proportional to the magnetic field component along the Néel vector, we attributed these two sample orientations as corresponding to the cases when the Néel vector remained parallel and perpendicular to the external magnetic field while increasing the magnetic field, respectively. Additionally, at the sample orientation corresponding to the maximum Zeeman splitting, we increased the magnetic field further to observe the spin-flop phase transition, which is highly sensitive to the angle between the magnetic field and the *a*-axis and can assist in more accurately determining the *a*-axis. A similar angle-resolved measurement has been used to determine the Néel vector at 0 T in the research of MnF₂¹².

By vertically polarizing the laser using a polarizer, we determined the polarization of the PL or laser when it propagated to the sample or to the spectrometer by introducing an analyzer or employing analysis based on the principle of light polarization. We measured the ZRE polarization while rotating the sample at 0 T and found that the polarization could synchronously follow the sample rotation, indicating that we can conveniently determine the *a*-axis through the polarization of ZRE at 0 T. To establish the relationship between the ZRE polarization and the *a*-axis at 0 T, we conducted the experiments as follows. According to the methods discussed above, we first rotated the sample to the orientation in which the *a*-axis is parallel with the magnetic field. Then, we measured the polarization of vertical polarized laser at 0 T as shown in Figure R1 **d** and compared it with the polarization of ZRE as shown in Figure R1 **e**. As a result, we then determined the polarization of ZRE is along the *a*-axis.

Benefited from the Zeeman splitting and polarization of the ZRE, we can identify the Néel vector and *a*-axis. In addition, we also used LD spectra to double check Néel vector and *a*-axis, as shown in Figure R1 **f**. A similar LD method has been used to

determine the Néel vector in NiPS_3 ^{9,13,14} and FePS_3 ^{14,15}.

Figure R1. **a**, The schematic setup of magneto-optical measurement system. **b**, Magnetic field-dependent Zeeman splitting and polarization of ZRE when the *a*-axis is along the magnetic field. **c**, Magnetic field-dependent Zeeman splitting and polarization of ZRE when the *a*-axis is perpendicular to the magnetic field. **d**, The measured polarization of the laser. **e**, The measured polarization of the ZRE at 0 T. **f**, LD measurement of the sample at 0 T.

Comments 3: In Figure 3, the Néel vectors are indicated. However, this is based on a rather simple model calculation. Since the direction of the Néel vectors and the exciton polarization is the main topic of this work, the direction should be determined from a separate measurement such as neutron scattering.

Reply: We appreciate the reviewer for raising this important question. As the reviewer mentioned, we simply extracted the orientation of Néel vector from the Zeeman splitting. We agree that a separate measurement to determine the Néel vector is helpful, while our sample is too small to be detected by the neutron scattering experiments. To make the determined Néel vector more convincing, we have conducted the Monte Carlo simulations, the temperature-dependent PL spectra measurements and the polarization-resolved and angle-resolved magneto-optical measurements from 0 T to 12T in this version.

As detailed in the SI note 3, we determine the direction of the Néel vector in NiPS_3 with the Monte Carlo simulations based on an XY-type Hamiltonian in a hexagonal lattice. The simulated spin orientations, depicted in the SI Figure S5, allowed us to calculate the orientation of Néel vector. These simulated results are in consistent with those extracted from the Zeeman splitting.

Furthermore, the temperature-dependent PL spectra, as shown in SI Figure S2a, confirmed the coupling of the X_1 with the antiferromagnetic order. Consequently, the Zeeman splitting of the X_1 exciton originates from the spins constructing the antiferromagnetic order and this simple calculation is reasonable.

As shown in the Figure 4e of the revised version, under various in-plane magnetic field directions and magnitudes, the Néel vector extracted from the Zeeman splitting has been verified to rotate synchronously with the polarization of the exciton, making the extraction more convincing.

Comments 4: In Figure 3, the experimental data agrees with the calculated curves only in the linear range. For example, the Zeeman splitting in Fig. 3b does not show the decrease with the magnetic field in the high-field range.

Reply: We appreciate the reviewer's inquiry. The primary reason for the discrepancy between the previous experimental data and the calculated curves is the larger error in the sample rotation angles. In prior experiments, reloading the sample was necessary to adjust the sample angles. In the new experiments, we utilized an in-situ rotator at low temperatures up to 12 T to change the sample angle with a smaller error below 3 degrees. The updated data in Figure 3 now shows that the simulated curves fit very well with the experimental data, even above the spin-flop transition, where the Zeeman splitting decreases with the magnetic field.

Comments 5: The argument for the 'spin-dipole separation' is not convincing. The only experimental observation is that the polarization of the exciton peak rotates with the in-plane magnetic field even when there is no Zeeman splitting. What does it mean that the spin and the dipole are separated? Does it mean that the exciton is separated from the magnetic ordering? If so, the temperature dependence of the exciton peak under in-plane magnetic field should be measured for different rotation angles.

Reply: We would like to express our gratitude to the reviewer for bringing up this significant point. As the reviewer rightly points out, the only experimental observation that the polarization of the exciton peak rotates with the in-plane magnetic field even when there is no Zeeman splitting is not sufficient to demonstrate the spin-dipole separation phenomenon. Another reviewer also suggested that the observation might be attributed to low spectral resolution or large errors in determining the exact 90° . Consequently, we reconducted experiments in the Voigt configuration. In comparison with the previous experiments, three improvements were implemented to enhance the accuracy of the experiments:

- 1) We used a higher-resolution spectrograph with an 1800 lines/mm grating to detect the Zeeman splitting.
- 2) We measured the polarization-resolved PL spectra and Zeeman splitting dependent on higher magnetic fields (up to 12 T) and obtained a more accurate evaluation of exciton polarization and Néel vector.
- 3) We used an in-situ rotary stepper positioner to control the relative angle more accurately between the a -axis and the magnetic field direction. Utilizing this in-situ rotator, we could more accurately rotate the Néel vector of the NiPS3

thin flake. Additionally, the NiPS3 thin flake was mechanically exfoliated and transferred onto the Si/SiO₂ substrate with gold-marker. The images of microscopic photographs can help us to validate the relative rotation of the sample, as illustrated in Figure R2.

With the more precise angular rotation of the sample, improved spectral resolution, and larger magnetic fields, we discovered that the Néel vector and the polarization of PL simultaneously rotated by the same angle, as shown in Figure 4 in the main text of the revised version, indicating that our proposed spin-dipole separation was not true. Especially, as shown in the Figure R2 **d**, **e** and **f**, we observed that the rotation of polarization of ZRE was rather tiny with the in-plane magnetic field when there was almost no Zeeman splitting in the latest experiments. As a result, we replaced the discussion about “spin-dipole separation” with “spin-dipole coupling” in the revised version. We sincerely apologize for the erroneous conclusion resulting from the lack of experimental precision.

Figure R2. Determination of relative sample orientation and the magnetic field-dependence of polarization. **a**, Photographs of gold-marker of the substrate with rotating the sample. The two arrows along the edges of the marker serve to identify the orientation of the marker. By treating the leftmost panel as an un-rotated sample orientation, i.e., rotated by 0 degrees, the relative rotation angles of the sample, as indicated in each panel, are extracted from the marker. **b**, Photographs of samples corresponding to the marker orientations in **a**. **c**, The extracted polarization of ZRE under different sample orientation without magnetic field from the polarization-resolved PL measurements. The dots represent PL intensity under different polarizer direction. The lines represent the fitted curves with a sine function. The angles indicated as “3°”, “-13°” and so on represent the fitted polarizer direction with the largest PL intensity. The meaning of the labels “Rotate 0°” and so on are the same as in **a**. **d**, polarization of the ZRE when the magnetic is almost perpendicular to the α -

axis, the rotation of the polarization is nearly zero. **e**, The contour plot of the magnetic field-dependent PL spectra of ZRE when the magnetic field is almost perpendicular with the *a*-axis. **f**, The extracted Zeeman splitting energy and polarization as functions of magnetic field.

References:

- 1 Fina, I. *et al.* Anisotropic magnetoresistance in an antiferromagnetic semiconductor. *Nat. Commun.* **5**, 4671 (2014).
- 2 Wang, H. *et al.* Giant anisotropic magnetoresistance and nonvolatile memory in canted antiferromagnet Sr₂IrO₄. *Nat. Commun.* **10**, 2280 (2019).
- 3 Ji, H. *et al.* Rotational symmetry breaking in superconducting nickelate Nd_{0.8}Sr_{0.2}NiO₂ films. *Nat. Commun.* **14**, 7155 (2023).
- 4 Wang, X. *et al.* Spin-induced linear polarization of photoluminescence in antiferromagnetic van der Waals crystals. *Nat. Mater.* **20**, 964-970 (2021).
- 5 Foner, S. High-Field Antiferromagnetic Resonance in MnF₂ Using Pulsed Fields and Millimeter Wavelengths. *Phy. Rev.* **107**, 683-685 (1957).
- 6 Tang, M. *et al.* Continuous manipulation of magnetic anisotropy in a van der Waals ferromagnet via electrical gating. *Nat. Electron.* **6**, 28-36 (2023).
- 7 Kriegner, D. *et al.* Multiple-stable anisotropic magnetoresistance memory in antiferromagnetic MnTe. *Nat. Commun.* **7**, 11623 (2016).
- 8 Wildes, A. R. *et al.* Magnetic structure of the quasi-two-dimensional antiferromagnet NiPS₃. *Phys. Rev. B* **92**, 224408 (2015).
- 9 Hwangbo, K. *et al.* Highly anisotropic excitons and multiple phonon bound states in a van der Waals antiferromagnetic insulator. *Nat. Nanotechnol.* **16**, 655-660 (2021).
- 10 Kang, S. *et al.* Coherent many-body exciton in van der Waals antiferromagnet NiPS₃. *Nature* **583**, 785-789 (2020).
- 11 Belvin, C. A. *et al.* Exciton-driven antiferromagnetic metal in a correlated van der Waals insulator. *Nat. Commun.* **12**, 4837 (2021).
- 12 King, A. R. & Rohrer, H. The spinflop bicritical point in MnF₂. *AIP Conf. Proc.* **29**, 420-421 (1976).
- 13 Tan, Q. *et al.* Revealing the three-state nematicity in atomically-thin antiferromagnetic NiPS₃ via magneto-optical effect. *arXiv: 2311.12201* (2023).
- 14 Zhang, Q. *et al.* Observation of giant optical linear dichroism in a zigzag antiferromagnet FePS₃. *Nano. Lett.* **21**, 6938-6945 (2021).
- 15 Zhang, H. *et al.* Cavity-enhanced linear dichroism in a van der Waals antiferromagnet. *Nat. Photon.* **16**, 311-317 (2022).

Reviewer #3 (Remarks to the Author):

The Zhang-Rice exciton (ZRE) in NiPS₃ is an emerging observation, but not fully understood yet. This work performed the magneto-optical measurements on NiPS₃ in a Voigt geometry, and observed the splitting of the ZRE under magnetic field ranging from 0 to 9 T. They further reported that the splitting is anisotropic depending on the direction of the applied B field. The mechanism behind the splitting is well discussed. But I also find some confusing points and some unconvincing conclusions in this work to be addressed before publishing on Nature Comm.

Reply: We appreciate the reviewer for the positive comments of our work and for giving many insightful suggestions to further improve our manuscript. Following the reviewer's valuable suggestions, we have done additional experiments and analysis, including:

- 1) Zeeman splitting as a function of the angle between the magnetic field and the Néel vector at a higher magnetic field up to 12 T,
- 2) Power-dependent PL measurement for the ZRE,
- 3) Linear dichroic (LD) measurement under zero magnetic field,
- 4) Polarization-dependent PL at high magnetic field up to 12 T,
- 5) Monte Carlo simulation of spin orientation of the NiPS₃ under a magnetic field ranging from 0 T to 12 T.

We have incorporated the findings of these experiments and analyses into our manuscript to enrich the content and improve its quality.

Comments 1: In the introduction, the authors said “There is no report about energy shift or splitting of the ZRE under magnetic field...” which is not true. This paper on arXiv reported the energy splitting of the exciton (<https://arxiv.org/pdf/2306.07660.pdf>), although there is still lack of understanding.

Reply: We thank the reviewer for pointing this issue out. The timing of our submission and the publication of the arXiv article were only two days apart. We learned about this arXiv article after our submission. Both works were completely conducted independently and contemporaneously. They mainly focus on the magnon gap excitations and spin-entangled optical transition in NiPS₃. It's worth noting that the anisotropic and nonlinear mechanism of the Zeeman splitting and polarization of the X₁ exciton, the temperature-dependence and origin of the X₁ exciton as well as the critical behavior of spin configuration above critical magnetic field were not extensively discussed in the arXiv article. In our research, we focus on the anisotropic Zeeman splitting and polarization of the ZRE in NiPS₃ under magnetic field. By changing the direction and magnitude of the in-plane magnetic field, we observed the anisotropic Zeeman splitting and polarization rotation of the ZRE, which are consistent with Monte Carlo simulation results. In the revised version, we have cited this paper in the end of the main text.

Comments 2: In the introduction, the authors said “... the PL linewidth of the exciton is broad (roughly 150 meV), which is not desirable for magneto-optical physics”. I am not sure if I agree with this. There are lots of things in magneto-optical physics to be

studied. I think there could be still interesting magneto-optical physics in broad excitons.

Reply: We agree with this view, and it is true that our statement is not rigorous. In revised version, we deleted this sentence.

Comments 3: The authors also mentioned that there are many controversies about the exciton properties in the introduction. For example, there are reports on the assignment of the exciton at 1.47 eV to both from the material itself and from the defects in the materials. It seems that the authors agree that this is an unanswered question, but then on page 3 (Figure 1d), the authors use the narrow bandwidth of the exciton to support the good crystallinity of the sample. The authors should at least give their evidence to support that the exciton is not from defects, so that they can make this conclusion.

Reply: We sincerely appreciate the reviewer's insightful suggestions. In order to exclude the defect-related origin of the X1 exciton in NiPS₃, power-dependent PL experiments were conducted, as shown in Figure R1 a, b and c. The PL intensity of the X1 exciton is proportional to the laser power and the PL peak energy is independent on the laser power. These phenomena can exclude the possibility that the PL originates from the defects, which agree well with referees^{1,2}. In addition, we also measured the PL by varying the exciting position of the sample as shown in Figure R1 d, all of which show the very similar behavior. In the revised version, we have also added these experiments in the SI Figure S3 and adjusted the corresponding paragraphs.

Figure R1. PL measurement of NiPS₃ under different laser power and sample position. **a**, Laser power-dependent PL spectra of ZRE excited by a 633 nm laser. **b**, Laser power-dependent PL intensity of ZRE. It shows a linear dependence of intensity on the excitation power in the range from 10 μW to 3 mW. **c**, The peak energy of ZRE shows little change under different excitation power. **d**, PL spectra measured in different excited positions of the same sample.

Comments 4: In addition to the X1 peak, the authors also observed another sharp peak

at 1.498 eV (X3). They cite reference 2 and 15 to say they are the two-magnon sideband. But there is another paper (Nature Materials volume 20, pages 964–970 (2021)) attributed it to a different origin. It would be better that the authors provide a more accurate picture to the readers on what the current understanding is.

Reply: We thank the reviewer for this insightful inquiry. For another sharp peak at 1.498 eV (X3), we did not research the origin of it. The high-energy one (labeled as X3) was earlier attributed to the two-magnon sideband or Rydberg exciton of the X1^{3,4}, while subsequent temperature-dependent experiments identified the X3 as the phonon sideband of the X1^{1,5}. We have adjusted the corresponding paragraph in the revised version.

Comments 5: The labels in the schematics in Figure 2 are not clear. For example, it is not indicated what is the red and blue balls, and what the arrows mean. The authors should be clear about the schematic presentation.

Reply: Thanks for pointing this out. We have carefully revised the legend in the revised version. The red and blue balls indicate the spin up and spin down of the Ni or S atom, while the small (large) balls represent the S (Ni) atom. The arrows mean the spin of the lattices.

Comments 6: The authors claim that they observed spin-dipole separation (Figure 4), which is not a convincing conclusion. They got this conclusion based on that there is no exciton splitting under B field with $\theta = 90^\circ$, but the polarization of the exciton rotates with B field along $\theta = 90^\circ$ direction. First, the reason they do not see the splitting could be because the spectra resolution is too low. Second, there could be a small error in determining the angle to be exact 90° . If we consider these two reasons, it will be hard to say that the spin and dipole are separate. In addition, the authors simply reported on the experimental observation (as I summarized above) without in depth understanding on why spin-dipole separation is reasonable in NiPS₃. In fact, there are already quite a few papers which are also cited by this paper, including some other results presented in this work, showed that the spin and the dipole are strongly coupled in NiPS₃. Thus, I am not convinced about the spin-dipole separation conclusion.

Reply: We appreciate the reviewer for the comments and suggestions. As the reviewer points out, the observation may be due to the spectral resolution is too low or there could be a small error in determining the angle to be exact 90° . Another reviewer also mentioned that the only experimental observation does not suffice to demonstrate the spin-dipole separation phenomenon. Therefore, we repeated the fluorescence experiments in the Voigt configuration. Compared with the previous experiments, three improvements were made to improve the accuracy of the experiments:

- 1) We used a higher-resolution spectrograph with an 1800 lines/mm grating to detect the Zeeman splitting.
- 2) We measured the polarized PL spectra and Zeeman splitting of ZRE dependent on higher magnetic fields (up to 12 T) and obtained a more accurate evaluation of magnetic field direction, exciton polarization and Néel vector.
- 3) We used an in-situ rotary stepper positioner to control the relative angle more

accurately between the Néel vector and the magnetic field direction. Utilizing the stepper positioner, we could more accurately rotate the Néel vector of the NiPS₃ thin flake. Additionally, the NiPS₃ thin flake was mechanically exfoliated and transferred onto the Si/SiO₂ substrate with gold-marker. The images of microscopic photographs can help us to validate the relative rotation of the sample, as illustrated in Figure R2 **a** and **b**.

With the more precise angular rotation of the sample, improved spectral resolution, and larger magnetic fields, we discovered that the Néel vector and the polarization of PL simultaneously rotate by the same angle as shown in the updated Figure 4. Especially, as shown in the Figure R2 **d**, **e** and **f**, we observed that the rotation of polarization of ZRE was rather tiny with the in-plane magnetic field when there was almost no Zeeman splitting in the latest experiments. The new data indicates that our proposed spin-dipole separation was not true. We sincerely apologize for the erroneous conclusion resulting from the lack of experimental precision.

Figure R2. Determination of relative sample orientation and the magnetic field-dependence of polarization. **a**, Photographs of gold-marker of the substrate with rotating the sample. The two arrows along the edges of the marker serve to identify the orientation of the marker. By treating the leftmost panel as an un-rotated sample orientation, i.e., rotated by 0 degrees, the relative rotation angles of the sample, as indicated in each panel, are extracted from the marker. **b**, Photographs of samples corresponding to the marker orientations in **a**. **c**, The extracted polarization of ZRE under different sample orientation without magnetic field from the polarization-resolved PL measurements. The dots represent PL intensity under different polarizer direction. The lines represent the fitted curves with a sine function. The angles indicated as “3°”, “-13°” and so on represent the fitted polarizer direction with the largest PL intensity. The meaning of the labels “Rotate 0°” and so on are the same as in **a**. **d**, polarization of the ZRE when the magnetic is almost perpendicular to the *a*-

axis, the rotation of the polarization is nearly zero. **e**, The contour plot of the magnetic field-dependent PL spectra of ZRE when the magnetic field is almost perpendicular with the *a*-axis. **f**, The extracted Zeeman splitting energy and polarization as functions of magnetic field.

Comments 7: In the method section, when describing the spin-dipole separation, it is not easy to follow as there are some unexplained terms in the math and missing references. For example, in "...only the C66 and C13 term survive..." what are C66 and C13? The authors should check the math in this work careful and make it easy to follow.

Reply: We appreciate the reviewer for raising this question. Since the polarization aligns along the Néel vector, we have removed the magnetic-electric coupling model from both the main text and the method. We are sorry for the mistake we have made.

Comments 8: How did the authors determine the *a*-axis of the crystal? One important conclusion the authors made is that the linear polarization of the exciton is along the *a*-axis of the crystal. But it is not clear how the authors determine the *a*-axis of the crystal.

Reply: We thank the reviewer for raising this question. We would like to provide the following discussions regarding the determination of the direction the *a*-axis.

Below the Néel temperature, the direction of the Néel vector of NiPS₃ is along the *a*-axis of the crystal, which has been confirmed by neutron scattering and X-ray diffraction experiments as shown in the reference⁶. This result has been widely accepted by many works such as references^{1,3,7,8}. Due to the small size of our sample, conducting X-ray or electron diffraction measurements on the same sample presents challenges. Therefore, we determine the *a*-axis by ascertaining the direction of the Néel vector with the application of an in-plane magnetic field. This is achieved through ZRE Zeeman splitting, ZRE polarization, and linear dichroism (LD) spectra for convenience.

The optical setup of our angle-resolved and polarization-resolved measurements is depicted in the Figure R3 **a**. We rotated the sample using an in-situ rotator while applying an in-plane magnetic field to the sample. In the angle-resolved PL experiments conducted from 0 T to 10 T, we observed that the Zeeman splitting of ZRE reached its maximum and exhibited high linearity in a specific orientation of the sample, as shown in Figure R3 **b**. Conversely, in the sample orientation perpendicular to this, the Zeeman splitting was almost negligible, as shown in the Figure R3 **c**. As the Zeeman splitting is proportional to the magnetic field component along the Néel vector, we attributed these two sample orientations as corresponding to the cases when the Néel vector remained parallel and perpendicular to the external magnetic field while increasing the magnetic field, respectively. Additionally, at the sample orientation corresponding to the maximum Zeeman splitting, we increased the magnetic field further to observe the spin-flop phase transition, which is highly sensitive to the angle between the magnetic field and the *a*-axis and can assist in more accurately determining the *a*-axis. A similar angle-resolved measurement has been used to determine the Néel vector at 0 T in the research of MnF₂⁹.

By vertically polarizing the laser using a polarizer, we determined the polarization of the PL or laser when it propagated to the sample or to the spectrometer by introducing

an analyzer or employing analysis based on the principle of light polarization. We measured the ZRE polarization while rotating the sample at 0 T and found that the polarization could synchronously follow the sample rotation, indicating that we can conveniently determine the a -axis through the polarization of ZRE at 0 T. To establish the relationship between the ZRE polarization and the a -axis at 0 T, we conducted the experiments as follows. According to the methods discussed above, we first rotated the sample to the orientation in which the a -axis is parallel with the magnetic field. Then, we measured the polarization of vertical polarized laser at 0 T as shown in Figure R3 d and compared it with the polarization of ZRE as shown in Figure R3 e. As a result, we then determined the polarization of ZRE is along the a -axis.

Benefited from the Zeeman splitting and polarization of the ZRE, we can identify the Néel vector and a -axis. In addition, we also used LD spectra to double check Néel vector and a -axis, as shown in the Figure R3 f. A similar LD method has been used to determine the Néel vector in NiPS₃^{7,10,11} and FePS₃^{11,12}.

Figure R3. **a**, Magneto-optical measurement system. **b**, Magnetic field-dependent Zeeman splitting and polarization of ZRE when the a -axis is along the magnetic field that range from 0 to 12 T. **c**, Magnetic field-dependent Zeeman splitting and polarization of ZRE when the a -axis is perpendicular to the magnetic field. **d**, The measured polarization of the laser. **e**, The measured polarization of ZRE at 0 T. **f**, LD measurement of the sample at 0 T.

Comments 9: The authors said that their results show we can use the exciton splitting for accurate spin read out. I understand the splitting is related to the angle between the Néel vector and the B field. But it would be better to provide another evidence from a difference angle to show that the Néel vector changed under B field, for example, a theoretical calculation or neutron scattering measurements (the neutron scattering

might be difficult to do on small samples). Also, as we know there should be a magnetic phase transition at $B > 10$ T, it would be better to show the data across the phase transition, so it would be more convincing.

Reply: We thank the reviewer for the comments and suggestions. We agree with the reviewer's suggestion. To confirm that the Néel vector is changed under the magnetic field, we measured the angle-resolved PL spectra of the ZRE at a high magnetic field up to 12 T, and conducted Monte Carlo simulations with a XY-type Hamiltonian on a hexagonal lattice.

The extracted Zeeman splitting and Néel vector from the angle-resolved PL spectra of the ZRE are depicted in Figure R4 **a** and **b**, which have been also updated in Figure 3 in the revised version. When the magnetic field is almost along the a -axis, a clear spin-flop transition is detected by the Zeeman splitting and is consistent with the PL polarization result in Ref¹. The Néel's formula could also fit the new data, as shown in the SI Figure S11.

Furthermore, we also performed Monte Carlo simulations to calculate the physical quantities in 1 layer of honeycomb lattice of spins. Each layer contains 16×16 honeycombs. The spin system is described by the Hamiltonian:

$$\mathcal{H} = \frac{1}{2} \sum_{\alpha=a,b,c} \left[\sum_{\langle i,j \rangle} J_{1\alpha} S_{i\alpha} S_{j\alpha} + \sum_{\langle\langle i,k \rangle\rangle} J_{2\alpha} S_{i\alpha} S_{k\alpha} + \sum_{\langle\langle\langle i,l \rangle\rangle\rangle} J_{3\alpha} S_{i\alpha} S_{l\alpha} \right] - D \sum_i (S_i^x)^2 + g\mu_B\mu_0 \mathbf{H} \cdot \mathbf{S}$$

where single, double, and triple angular brackets in the sums denote the nearest, the next-nearest, and the third-nearest neighbors on the same plane, respectively. $J_{1\alpha}, J_{2\alpha}, J_{3\alpha}$ are the nearest, the next-nearest, the third-nearest coupling parameters along α direction, respectively. D is easy-axis single-ion anisotropy. In our simulation, we set the parameters along the different direction to be the same and $(J_{1\alpha}, J_{2\alpha}, J_{3\alpha}, D)$ to be $(-1.9, 0.1, 1.9, 0.08)$ eV. The results are shown in Figure R3 **c** and **d**, and we also added this part in the SI Figure S5. Figure R3 **d** shows that the orientation of Néel vector calculated from Monte Carlo simulation is consistent with our experimental results.

Finally, we also plotted the PL polarization of ZRE as a function of the magnetic field, as shown in Figure R5, comparing the results of different angle at 9 T and 12 T, the rotation of the Néel vector detected by the PL polarization and the Zeeman splitting is the same.

Figure R4. Angle-dependent Zeeman splitting of ZRS exciton. **a**, The Zeeman splitting energy as a function of the angle θ and magnetic field. The inset cartoon schematically illustrates the angle θ between magnetic field and the a -axis. The dots are the experiment data, while the lines are the fit curves. **b**, The extracted φ from **a**. **c**, Monte Carlo simulation results of field dependence of Zeeman splitting. **d**, Monte Carlo simulation results of field dependence of φ .

Figure R5. PL polarization rotation of the ZRE as a function of the magnetic field. a, The scattering points are measured polarizations and the curves are the same as curves in Figure R3 **c. b,** Anisotropic rotation of the polarization of ZRE and the Néel vector when the magnetic field is 9 T and 12 T. The dash and solid lines are calculated from Néel's formula.

References:

- 1 Wang, X. *et al.* Spin-induced linear polarization of photoluminescence in antiferromagnetic van der Waals crystals. *Nat. Mater.* **20**, 964-970 (2021).
- 2 Kim, J. *et al.* Rapid Suppression of Quantum Many-Body Magnetic Exciton in Doped van der Waals Antiferromagnet (Ni,Cd)PS₃. *Nano Lett.* **23**, 10189-10195 (2023).
- 3 Kang, S. *et al.* Coherent many-body exciton in van der Waals antiferromagnet NiPS₃. *Nature* **583**, 785-789 (2020).
- 4 Ho, C.-H., Hsu, T.-Y. & Muhimmah, L. C. The band-edge excitons observed in few-layer NiPS₃. *npj 2D Mater. Appl.* **5**, 8 (2021).
- 5 Wang, X. *et al.* Electronic Raman scattering in the 2D antiferromagnet NiPS₃. *Sci. Adv.* **8**, eabl7707 (2022).
- 6 Wildes, A. R. *et al.* Magnetic structure of the quasi-two-dimensional antiferromagnet NiPS₃. *Phys. Rev. B* **92**, 224408 (2015).
- 7 Hwangbo, K. *et al.* Highly anisotropic excitons and multiple phonon bound states in a van der Waals antiferromagnetic insulator. *Nat. Nanotechnol.* **16**, 655-660 (2021).
- 8 Belvin, C. A. *et al.* Exciton-driven antiferromagnetic metal in a correlated van der Waals insulator. *Nat. Commun.* **12**, 4837 (2021).
- 9 King, A. R. & Rohrer, H. The spinflop bicritical point in MnF₂. *AIP Conf. Proc.* **29**, 420-421 (1976).
- 10 Tan, Q. *et al.* Revealing the three-state nematicity in atomically-thin antiferromagnetic NiPS₃ via magneto-optical effect. *arXiv: 2311.12201* (2023).
- 11 Zhang, Q. *et al.* Observation of giant optical linear dichroism in a zigzag antiferromagnet FePS₃. *Nano. Lett.* **21**, 6938-6945 (2021).
- 12 Zhang, H. *et al.* Cavity-enhanced linear dichroism in a van der Waals antiferromagnet. *Nat. Photon.* **16**, 311-317 (2022).

Reviewers' Comments:

Reviewer #2:

Remarks to the Author:

The authors revised the manuscript extensively with additional data. Most of the previous concerns have been addressed. Although some of the results (polarization rotation of photoluminescence) still have questions of novelty, the new results on Zeeman splitting and the asymmetric angle dependence of the Neel vector under magnetic field are interesting enough. I think this work can be published in Nature Communications. I have no further comments.

Reviewer #4:

Remarks to the Author:

This manuscript demonstrates the excitonic Zeeman splitting in NiPS₃. The many-body excitons in NiPS₃ with ultrasharp linewidth are intriguing phenomena. Previous publications already show these sharp excitons and their linear polarization strongly interact with the AFM order. The current paper confirms that the exciton dipole is aligned with the local spin through magneto-PL measurements across a spin-flop transition.

The exciton Zeeman measurements in this paper are clean and clear, however, the paper does not offer sufficient new insights on the nature of Zhang-Rice excitons by the Nat Comm. standard. While the authors have provided the phenomenological fitting of exciton energy and polarization during the spin-flop process, there is one essential point that needs to be addressed. It is important to elucidate how the dipole of the on-site excitons aligns with the local spin with the help of spin-orbit coupling. I strongly suggest the authors provide further discussions and calculations on this point, especially from a microscopic viewpoint

Furthermore, the claim of multiple-stable spin configurations or multi-critical behaviors in Figure 5 is questionable. The author has the assumption of monodomain during the spin-flop process, which may not be the case. Inhomogeneity can easily give rise to domains with different spin-flop fields within the beam spot. Therefore, further evidence must be provided if the authors insist on claiming the multi-critical behavior and high-order anisotropy effects.

In addition, the schematic diagram of Figure 4a is a little confusing. What does the hexagon represent here? the unit cell? The Zhang-Rice exciton is on-site, however, the exciton dipole in the figure looks like an inter-site or a Wannier exciton with a large

dipole displacement.

Point-to-point response to the referees' comments.

Reviewer #2:

The authors revised the manuscript extensively with additional data. Most of the previous concerns have been addressed. Although some of the results (polarization rotation of photoluminescence) still have questions of novelty, the new results on Zeeman splitting and the asymmetric angle dependence of the Neel vector under magnetic field are interesting enough. I think this work can be published in Nature Communications. I have no further comments.

Reply: We sincerely appreciate the reviewer for his/her positive comments on the innovation of our work, as well as his/her careful review of our manuscript and the valuable suggestions and comments he/she has provided.

Reviewer #4:

Comments #1:

This manuscript demonstrates the excitonic Zeeman splitting in NiPS3. The many-body excitons in NiPS3 with ultrasharp linewidth are intriguing phenomena. Previous publications already show these sharp excitons and their linear polarization strongly interact with the AFM order. The current paper confirms that the exciton dipole is aligned with the local spin through magneto-PL measurements across a spin-flop transition.

Reply #1:

We appreciate the thorough review and insightful suggestions provided by the reviewer. As the review mentioned, the many-body excitons in NiPS3 with ultrasharp linewidth are indeed intriguing phenomena. Our measurements reveal highly anisotropic Zeeman splitting of the excitons, which can reflect multi-stable spin configurations under a magnetic field. We believe these phenomena are novel and significant for this field. We will address the reviewer's comments point by point, particularly focusing on explaining the alignment of local spin and dipole of ZRE based on spin-orbit coupling.

Comments #2:

The exciton Zeeman measurements in this paper are clean and clear, however, the paper does not offer sufficient new insights on the nature of Zhang-Rice excitons by the Nat Comm. standard. While the authors have provided the phenomenological fitting of exciton energy and polarization during the spin-flop process, there is one essential point that needs to be addressed. It is important to elucidate how the dipole of the on-site excitons aligns with the local spin with the help of spin-orbit coupling. I strongly suggest the authors provide further discussions and calculations on this point, especially from a microscopic viewpoint.

Reply #2:

We appreciate the reviewer for the positive evaluation of our Zeeman measurement. The observed anisotropic Zeeman effect, as well as the spin-dipole alignment and the multiple spin configurations, are novel and intriguing. As Referee #2 said, “The new results on Zeeman splitting and the asymmetric angle dependence of the Neel vector under magnetic field are interesting enough.” Furthermore, we present a new understanding about the 2D magnetic phase transition from these phenomena.

The role played by ZRE, as highlighted by the reviewer, is indeed significant. Following the referee’s suggestion, we will provide a more detailed explanation of the origin of the Zeeman effect by determining the photoluminescence (PL) as the π light^{1,2}, then elucidating the alignment of the spin and dipole moments in ZRE. This explanation will also address the long-standing contradictions concerning the short lifetime of ZRE.

In the Zeeman effect in metal ions transitions, as shown in Figure R1a, linearly polarized light collected parallel to the magnetic field is referred to as π light^{1,2}. Here π light corresponds to transitions with unchanged magnetic quantum numbers, meaning the angular momentum along the magnetic field direction remains the same during the transition. As a result, the propagating direction of the emitted photons is perpendicular to the magnetic field. The probability of photon propagation in all directions within the plane perpendicular to the magnetic field is equal, which cancels out the electric field component perpendicular to the magnetic field, leaving only the electric field component parallel to the angular momentum, resulting in the polarization of π light being parallel to the magnetic field. Even if the magnetic field is rotated, the π light rotates accordingly and always keeps parallel to the magnetic field.

In our experiment, as shown in Figure R1b and c, the orientation of local magnetic moments was identified using anisotropic Zeeman splitting, and it was found that the polarization of PL remained parallel to the local magnetic moments in all orientations. As a direct result of our polarization experiment, the PL is π light and the total magnetic quantum number remains unchanged during the radiative transition of spin-correlated excitons. The spin-correlated exciton corresponds to a spin-flip process^{3,4}, implying that the component of orbital angular momentum in the spin direction should change to preserve total angular momentum. This orbital change is allowed by that both Zhang-Rice singlets and triplets compose by a d -orbital hole and a p -orbital hole⁵, excluding the potential d - d transition mechanism.

Within NiPS3, the Zhang-Rice singlet is located in an octahedron consisting of a Ni atom and six S atoms, where a hole occupies a d orbital of the Ni atom, and a portion of the p orbitals of the six S atoms in the octahedron contribute to the hole³. According to the theory proposed by Zhang and Rice, the ZRS orbitals form bonding states with strong overlap between the ligands and the transition metal⁶. In ZRS, the symmetry of the hole on the ligands matches that of the hole on the transition metal, leading to the

consideration of the hole on the S atoms as a d orbital hole. On the other hand, the ZRT orbitals form antibonding states, resulting in relatively independent ligand and transition metal holes, with the hole on the S atoms maintaining its original p orbital characteristics⁷. Even though there is orbital quenching, the p orbitals located outside the easy plane can still contribute to orbital angular momentum⁷. Therefore, the transition from ZRS to ZRT causes a change in orbital angular momentum since it corresponds to the transition from a d orbital hole to a p orbital hole.

The orbital change helps explain why the ZRE has a shorter lifetime compared to optical transitions in other $3d$ ions, such as CuB_2O_4 ⁸, MnF_2 ⁹, and Cr_2O_3 ¹⁰, which arise from $d-d$ transitions that are parity-forbidden. In contrast, the change in orbital angular momentum means ZRE is allowed by the dipole transitions selection rule, leading to larger transition matrix element and then rapider radiation transition rate and shorter lifetimes¹¹. Additionally, the involvement of p orbitals of S atom introduces $J-J$ coupling^{12,13}, avoiding spin-forbidden limit and increasing the radiation transition rate compared with spin flip process in $d-d$ transition.

As shown in Figure R1d, this orbital behavior complements our understanding of spin reorientation transition, where the magnetic field reorientate the local magnetic moments $\vec{\mu}$ to a new stable direction in NiPS_3 . Under this field, ZRT and ZRS possess different gyromagnetic ratios g , resulting in anisotropic Zeeman splitting. The total local magnetic moments originate from the spin angular momentum and orbital angular momentum. Since the orientation of orbital angular momentum is related to the orientation of orbital wave function, the magnetic field simultaneously drives the rotation of both spin and orbital. Due to orbital reduction effect^{14,15}, the g -factor contribution from orbital is reduced and the g factor of ZRS is small, while the main contribution to g factor of ZRT comes from the spin and is big. As a simplification, it is assumed that the spin-flip occurs in the central d orbitals, while the orbital change occurs in the p orbitals and the p hole transition from ${}^2D_{5/2}$ to ${}^2P_{3/2}$. The $p-d$ orbital hybridization could reduce the orbital g factor and the orbital reduction factor is denoted as k . The magnetic moment of ZRS could be calculated to be $0.8k_{ZRS}\mu_B$ considering the orbital contribution from ${}^2D_{5/2}$ state in the $L-S$ coupling scheme¹⁵. The magnetic moment of ZRT mostly originates from the total spin ($2\mu_B$) and the orbital contribution ($0.6k_{ZRT}\mu_B$) of ${}^2P_{3/2}$. The difference of local magnetic moment between ZRS and ZRT could be calculated to be $(2 + 0.8k_{ZRS} - 0.6k_{ZRT})\mu_B$. As discussed above the $p-d$ orbital hybridization is weaker in ZRT, k_{ZRS} is smaller than k_{ZRT} . This is consistent with the experimentally measured splitting energy of $3.9\mu_B B$ when a magnetic field is parallel with the local magnetic moment. The accurate g -factor for the ZRT in NiPS_3 can be obtained through electron paramagnetic resonance measurements¹⁵. Our Zeeman result could help obtain the g -factor and k -factor of the excited state and then help specifically determine the orbital configuration of the ZRS¹⁵.

In summary, the polarization direction and spin-polarization alignment of ZRE suggest that the change in orbital compensates for the change in angular momentum during the spin-flip transition from ZRS to ZRT. Under the conservation of angular momentum, the rotation of exciton polarization originates from the rotation of local magnetic moments, corresponding to synchronous rotations of both spin and orbital of ZRE. The change in orbital angular momentum could be attributed to the alteration in the exchange symmetry of orbital accompanying the spin-flip. The orbital change, allowing the selection rule of electric dipole transition and then obtaining rapider radiation transition rate, could explain the short lifetime of the spin-correlated excitons. The orbital change could exclude the potential d-d transition interpretation and then support the Zhang-Rice interpretation of this exciton.

Figure R1. Polarization, magnetic moment, and Zeeman effect of ZRE. (a) Polarization of Zeeman effect in metal ions. (b) Polarization of ZRE in our experiment. The blue and red arrow denote the local magnetic moment at Ni site in the lattice. (c) Cartoon of the polarization rotation of ZRE. The β is defined as the rotation angle of polarization of ZRE with and without magnetic field. The blue, red, and black arrows denote the local magnetic moment at Ni site in the lattice, the polarization of the exciton localized around one Ni site, and the external magnetic field, respectively. The dash and solid

arrow denote the condition with and without the external field. (d) The transition between ZRT and ZRS under the external field.

Comments #3:

Furthermore, the claim of multiple-stable spin configurations or multi-critical behaviors in Figure 5 is questionable. The author has the assumption of monodomain during the spin-flop process, which may not be the case. Inhomogeneity can easily give rise to domains with different spin-flop fields within the beam spot. Therefore, further evidence must be provided if the authors insist on claiming the multi-critical behavior and high-order anisotropy effects.

Reply #3:

For the multi-stable spin configurations hypothesized in Figure 5, we present additional evidence to prove monodomain.

In Figure R2a and c, NiPS₃ is supposed to possess three domain orientations with Néel vectors separated by angles of 120 degrees¹⁶. Therefore, when a magnetic field is inclined at an angle θ with one domain, the corresponding angles with the other two domains are $120^\circ - \theta$ and $120^\circ + \theta$. As depicted in Figure R2b and d, scenarios where the magnetic field is oriented at 0 degrees and 15 degrees with one domain are shown. It can be observed that under different polarization configurations, various sets of Zeeman splitting are present. By fitting the magnetic phase transition model and parameters as discussed in the main text, the Néel vector orientations and angles with the magnetic field corresponding to different splits are determined. The spectra in Figure R2b and d are in complete agreement with the multi-domain model represented by Figure R2a and c, indicating that under low fields, completely different sets of Zeeman splitting are observed in different polarization configurations in the multi-domain samples. In the samples used in our main text, different polarization configurations and sample angles under magnetic fields below 9 T consistently exhibit only one set of Zeeman splitting peaks, thus excluding the presence of multiple domains in the samples in the low field.

Regarding the case that new domains are induced beyond the critical field and the coexistence of multiple domains results in the appearance of different critical fields, we will exclude this case based on Figure 5 in the main text. Due to the lattice's three-fold rotary symmetry, the new domain should be oriented at 120 degrees from the original domain. Consequently, the splitting size of the new spectral lines should be completely distinct from the original lines. However, the spectra from 9 T to 11 T displayed in Figure 5 show that the splitting size of the new lines gradually moves away from the original lines. Therefore, the coexistence of multiple domains could be excluded.

Figure 2 . PL spectra of ZRE with the coexistence of multiple domains. (a) Cartoon of the domain states when the magnetic field is parallel with L_1 . (b) Spectra of the domain L_1 when the polarizer is parallel to L_1 . (c) Spectra of the domain L_2 and L_3 when the polarizer is perpendicular to L_1 . (d) Cartoon of the domain state when the angle between the magnetic field and L_1 is 15° . (e) Spectra of the domain L_1 when the polarizer is parallel to L_1 . (f) Spectra of the domain L_2 and L_3 when the polarizer is perpendicular to L_1 .

Comments #4:

In addition, the schematic diagram of Figure 4a is a little confusing. What does the hexagon represent here? the unit cell? The Zhang-Rice exciton is on-site, however, the exciton dipole in the figure looks like an inter-site or a Wannier exciton with a large dipole displacement.

Reply #4:

Regarding the unclear depiction of the hexagonal symbol in Figure 4a, we have clarified the description in the revised figure legend and distinctly illustrated the relationship between the exciton dipole moment and the local magnetic moment.

Figure R3. Cartoon of the polarization rotation of ZRE. The β is defined as the rotation angle of polarization of ZRE with and without magnetic field. The blue, red, and black arrows denote the local magnetic moment at Ni site in the lattice, the polarization of the exciton localized around one Ni site, and the external magnetic field, respectively. The dash and solid arrow denote the condition

with and without the external field.

Reference

- 1 Andelkovic, Z. *et al.* Laser cooling of externally produced Mg ions in a Penning trap for sympathetic cooling of highly charged ions. *Phys. Rev. A* **87**, 033423 (2013).
- 2 Wang, G. *et al.* In-Plane Propagation of Light in Transition Metal Dichalcogenide Monolayers: Optical Selection Rules. *Phys. Rev. Lett.* **119**, 047401 (2017).
- 3 Kang, S. *et al.* Coherent many-body exciton in van der Waals antiferromagnet NiPS₃. *Nature* **583**, 785-789 (2020).
- 4 Klaproth, T. *et al.* Origin of the Magnetic Exciton in the van der Waals Antiferromagnet NiPS₃. *Phys. Rev. Lett.* **131**, 256504 (2023).
- 5 Lane, C. & Zhu, J.-X. Thickness dependence of electronic structure and optical properties of acorrelated van der Waals antiferromagnetic NiPS₃ thin film. *Phys. Rev. B* **102** (2020).
- 6 Zhang, F. C. & Rice, T. M. Effective Hamiltonian for the superconducting Cu oxides. *Phys. Rev. B* **37**, 3759-3761 (1988).
- 7 J. Stöhr and H. C. Siegmann, Magnetism: From Fundamentals to Nanoscale Dynamics. *Springer, Heidelberg* **7**, 224-310 (2006).
- 8 Kudlacik, D. *et al.* Exciton and exciton-magnon photoluminescence in the antiferromagnet CuB₂O₄. *Phys. Rev. B* **102** (2020).
- 9 Sell, D. D., Greene, R. L. & White, R. M. Optical Exciton-Magnon Absorption in MnF₂. *Phys. Rev.* **158**, 489-510 (1967).
- 10 van der Ziel, J. P. Optical Spectrum of Antiferromagnetic Cr₂O₃. *Phys. Rev.* **161**, 483-492 (1967).
- 11 Mizushima, M. & Koide, S. On the Lifetime of the Lower Triplet States of Benzene. *The Journal of Chemical Physics* **20**, 765-769 (1952).
- 12 D. Cowan, The Theory of Atomic Structure and Spectra. *University of California Press, Berkeley* (1981).
- 13 J. Stöhr and H. C. Siegmann, Magnetism: From Fundamentals to Nanoscale Dynamics. *Springer, Heidelberg* **9**, 418 (2006).
- 14 Chai, R.-P., Kuang, X.-Y., Zhang, C.-X., Duan, M.-L. & Wang, H. Theoretical study of EPR spectra and local structure for (NiO₆)₁₀- cluster in LiNbO₃:Ni²⁺ and Al₂O₃:Ni²⁺ systems. *Journal of Physics and Chemistry of Solids* **69**, 1848-1854 (2008).
- 15 Gerloch M, Miller J R. Covalence and the orbital reduction factor, k, in magnetochemistry. *Progress in Inorganic Chemistry*, 1-47 (1986).
- 16 Tan, Q. *et al.* Revealing the three-state nematicity in atomically-thin antiferromagnetic NiPS₃ via magneto-optical effect. *arXiv: 2311.12201* (2023).

Reviewers' Comments:

Reviewer #4:

Remarks to the Author:

The rebuttal letter and the revised manuscript addressed some of the questions. The remaining questions and comments are the following:

1. The revised manuscript states “provide an explanation for the *long* lifetime of ZRE”(3rd paragraph, highlighted) In the latter part of the paper, authors try to explain the *short* lifetime of ZRE due to p-d transition. This is inconsistent. In any case, please comment on the linewidth and the lifetime of the ZRE. What is the limiting factor of the exciton lifetime, is the PL lineshaped inhomogeneously broadened or not?

2. Regarding the claim of ‘multiple-stable spin configurations’ and ‘multi-critical behaviors’. The reply #3 is confusing and fails to address the question. It seems the author considers the multi-domain as 3-state zigzag AFM domains. Actually, here I mean local inhomogeneity-induced domains with slightly different spin-flop fields, which is an extrinsic effect and has nothing to do with high-order anisotropy. To me, none of the evidence shown in the manuscript can exclude this simple inhomogeneity scenario. Therefore, it is confusing and groundless to use the terms ‘multiple-stable’ and ‘multi-critical’.

Point-to-point response to the referees' comments.

Reviewer #4

Comments #1 The rebuttal letter and the revised manuscript addressed some of the questions. The remaining questions and comments are the following:

1. The revised manuscript states, “provide an explanation for the long lifetime of ZRE” (3rd paragraph, highlighted). In the latter part of the paper, authors try to explain the short lifetime of ZRE due to p-d transition. This is inconsistent. In any case, please comment on the linewidth and the lifetime of the ZRE. What is the limiting factor of the exciton lifetime, is the PL lineshaped inhomogeneously broadened or not?

Reply #1:

We appreciate the reviewer pointing out the inconsistency and providing important suggestions. The use of "long" in the third paragraph in the main manuscript was a typo and should be "short". This has been corrected in the current version. The lifetime and linewidth of ZRE indeed warrant a more detailed discussion. To make the discussion more reliable, we measured the lifetime of the ZRE, as shown in Figure R1. In table R1, we also listed the reported lifetime and the linewidth of the exciton emission in NiPS₃. You can see that all samples synthesized by the chemical vapor transport (CVT) method have a short lifetime from 10 ps to 40 ps and narrow linewidth from 260 to 770 μeV , but the sample synthesized by liquid phase exfoliation (LQE) method has a longer lifetime of 1 ns and broader linewidth of 1.7 meV. In order to measure the lifetime, we re-prepared the samples using the mechanical exfoliation method. The sample shows ~ 17 ps lifetime and ~ 482 μeV linewidth from Figure R1, showing a comparable crystal quality with other reports that use CVT samples.

The experimentally measured linewidth and lifetime of ZRE correspond to the decay of different physical quantities. The mentioned ZRE lifetime in the main text corresponds to the exciton population decay time (T_1), which includes contributions from both radiative lifetime (T_{rad}) and non-radiative lifetime ($T_{\text{non-rad}}$) [1, 2]: $\frac{1}{T_1} =$

$\frac{1}{T_{\text{rad}}} + \frac{1}{T_{\text{non-rad}}}$. The linewidth (Γ) of the exciton peak corresponds to the $\frac{\hbar}{T_2}$, which is related to the overall phase relaxation time T_2 of the excitons [1, 2]. The Γ is not only from the exciton population decay rate ($\sim 1/2T_1$), but also from the pure dephasing rate ($\sim 1/T_2^*$) and inhomogeneous broadening (Γ_{inhom}). The pure dephasing of the exciton includes contributions from scattering with phonons, other electron excitations, and defects [2]. The relationship among them is given by [1,2]: $\Gamma = \frac{\hbar}{T_2} \sim \frac{\hbar}{2T_1} + \frac{\hbar}{T_2^*} + \Gamma_{\text{inhom}}$.

The population decay and pure dephasing processes ($\frac{\hbar}{2T_1} + \frac{\hbar}{T_2^*}$) contribute to homogeneous broadening.

At low temperature, the Γ of ZRE in references [3-6] and this work is generally larger than $\frac{\hbar}{2T_1}$, indicating that the mechanisms dominating the ZRE linewidth are pure dephasing or inhomogeneous broadening [3]. The linewidth of ZRE in LQE samples is larger than in CVT-grown samples, which can be attributed to the increased disorder in LQE leading to greater inhomogeneous broadening and disorder-related pure dephasing [5]. The temperature and laser power-dependent PL help us to understand the pure dephasing contributions to line broadening associated with phonons and other electron excitations [2,9]. As shown in Figure R2, the linewidth of ZRE remains almost unchanged when varying temperature below 20 K and varying laser power at 4 K. This suggests that at 4 K, phonons or other electronic excitations hardly contribute to the linewidth broadening of ZRE. The narrower linewidth of ZRE suggests that defect-related broadening of ZRE is smaller, which could originate from its spin-correlated nature and BEC-like coherence [7].

For the ZRE lifetime, the effect of inhomogeneous broadening mainly increases disorder and then increases the non-radiative transition rate in principle [2]. However, the observation that increasing disorder extends the ZRE lifetime [5] suggests that disorder suppresses radiative transition and the radiative transition dominates the ZRE lifetime. Compared with other excitons, the shorter lifetime of ZRE originates from the faster radiative transition. Our polarization experiments reveal that the ZRE transition process follows the orbital selection rules, which, to some extent, explains the higher radiative transition rate (i.e., shorter lifetime) of ZRE.

In summary, the narrow linewidth of the ZRE could originate from small inhomogeneous broadening and small defect-related pure dephasing due to its spin-correlated nature and BEC-like coherence [7]. The short lifetime of ZRE is due to its faster radiative transition rate, while the inhomogeneous broadening effect inhibits radiative transitions, thereby extending the ZRE lifetime in some samples such as liquid phase exfoliated samples.

Figure R1. (a), The PL and its fitting of ZRE. (b), The time-resolved PL spectrum of ZRE and its fitting. The time-resolved PL experiment was conducted using a time-correlated single photon counting technique and was excited by 440 nm pulse laser at 4 K. We obtain the linewidth (Γ) using

the following equation: $\text{FWHM}_{exp} = \sqrt{\Gamma^2 + \Gamma_{inst}^2}$, where FWHM_{exp} is the experimentally measured full-width-of-half-maximum of ZRE, Γ_{inst}^2 is instrument broadening, respectively. In our sample, a typical measured linewidth is $\text{FWHM}_{exp} = 533 \mu\text{eV}$, Γ_{inst} is $226 \mu\text{eV}$ and we got $\Gamma = 482 \mu\text{eV}$. The narrowest linewidth we measured is $260 \mu\text{eV}$ and corresponding $\Gamma = 128 \mu\text{eV}$. As shown in (b), we got the lifetime $T_1 = 17 \text{ ps}$ by deconvoluting the time-resolved PL spectra. The instrument response function (IRF) is 20 ps . In principle, our system has a time resolution of $\text{IRF}/5$.

Figure R2. (a), The temperature-dependent FWHM of ZRE, extracted from Supplementary Figure S2c. (b), The laser power-dependent FWHM of ZRE at 4K, extracted from Supplementary Figure S3d.

	Lifetime (T_1)	$\hbar/2T_1$	Linewidth	Samples
Ref. 3	11 ps @ 5 K	30 μeV	350 μeV @ 10 K	CVT
Ref. 4	2~9 ps @ 10 K	36 μeV ~164 μeV	330 μeV @ 5 K	CVT
Ref. 5	40 ps @ 10 K	8 μeV	770 μeV @ 10 K	CVT
Ref. 6	~1 ns @ 4K	~0.3 μeV	1.7 meV @ 4 K	LQE
This work	17 ps @ 4K	19 μeV	482 μeV @ 4 K	CVT

Table R1. The reports about the linewidth and lifetime of the ZRE. CVT means chemical vapor transport and LQE means liquid phase exfoliation.

Comments #2:

- Regarding the claim of ‘multiple-stable spin configurations’ and ‘multi-critical behaviors’. The reply #3 is confusing and fails to address the question. It seems the author considers the multi-domain as 3-state zigzag AFM domains. Actually, here I mean local inhomogeneity-induced domains with slightly different spin-flop fields, which is an extrinsic effect and has nothing to do with high-order anisotropy. To me,

none of the evidence shown in the manuscript can exclude this simple inhomogeneity scenario. Therefore, it is confusing and groundless to use the terms ‘multiple-stable’ and ‘multi-critical’.

Reply #1:

We appreciate the reviewer's patience in reinterpreting the meaning of "domain" and pointing out the confusing areas. Indeed, we understood domains to be 3-state zigzag AFM domains.

The phenomenon near the critical field in the main text Figure 5 is interesting, reflecting the non-trivial change of magnetic order in the transition region near the critical field, and provides good experimental results for understanding the magnetic structural evolution during phase transitions. Regarding the question of whether this phenomenon is multi-critical or not, we agree that it remains to be further investigated. We agree with the reviewer that the inhomogeneity mechanism could be a more direct explanation for this phenomenon. We notice that the research on the inhomogeneity effect could contribute to the domain engineering of antiferromagnetic (AFM) material and then provide potential application in AFM spintronics ^[10]. Therefore, the observation in Figure 5 is important and interesting. Based on the suggestion of the reviewer, we make modifications to the last paragraph of the discussion section in the main text, primarily including the following points:

The multiple peaks reflect the change of magnetic order in the transition region near the critical field. This phenomenon could originate from inhomogeneity-induced various spin-flop critical fields ^[12] or the transition to an intermediate phase ^[8]. If the phenomenon originates from inhomogeneity, our findings can help to further investigate the origin of inhomogeneity and thus contribute to the domain engineering in antiferromagnetic (AFM) materials ^[10]. If the phenomenon originates from the transition to an intermediate phase, the neutron scattering could help to further investigation ^[8] and the magnetic field dependence of the peak positions and polarization could contribute to the study of magnetic phase transitions in XXZ-type or XY-type AFM ^[11]. Further determination of the origin needs more experimental evidence and is beyond the scope of our discussion. Our results show that the PL polarization and energy of ZRE is a non-destructive and convenient method to detect the critical behaviors of NiPS₃.

Furthermore, due to the uncertain underlying mechanisms of this critical phenomenon, we remove sections from both the main text and the Supplementary Information that discussed various spin configurations induced by multiple types of anisotropy. More respective revisions are highlighted in the updated version.

Reference

- [1] Stavrias, N. et al. Competition between homogeneous and inhomogeneous broadening of orbital transitions in Si:Bi. *Phys. Rev. B* 96, 155204 (2017).
- [2] Klingshirn, C. *Semiconductor optics*. Springer, Berlin, Heidelberg 623-700 (2012).

- [3] Hwangbo, K., Zhang, Q., Jiang, Q. et al. Highly anisotropic excitons and multiple phonon bound states in a van der Waals antiferromagnetic insulator. *Nat. Nanotechnol.* 16, 655 – 660 (2021).
- [4] Wang, X., Cao, J., Lu, Z. et al. Spin-induced linear polarization of photoluminescence in antiferromagnetic van der Waals crystals. *Nat. Mater.* 20, 964–970 (2021).
- [5] Shcherbakov, A. et al. Solution-processed NiPS₃ thin films from liquid exfoliated inks with long-lived spin-entangled excitons. *ACS Nano* 17, 10423–10430 (2023).
- [6] Li, Y., Liang, G., Kong, C., Sun, B. & Zhang, X. Charge-transfer-mediated exciton dynamics in van der Waals antiferromagnet NiPS₃. *Adv. Funct. Mater.* n/a, 2402161.
- [7] Kang, S., Kim, K., Kim, B.H. et al. Coherent many-body exciton in van der Waals antiferromagnet NiPS₃. *Nature* 583, 785–789 (2020).
- [8] Smeets, J. P. M., Frikkee, E. & de Jonge, W. J. M. Intermediate phase in a spin-flop system with coupled order parameters. *Phys. Rev. Lett.* 49, 1515–1518 (1982).
- [9] Moody, G., Kavir Dass, C., Hao, K. et al. Intrinsic homogeneous linewidth and broadening mechanisms of excitons in monolayer transition metal dichalcogenides. *Nat. Commun.* 6, 8315 (2015).
- [10] Thong, H.-C. et al. Domain engineering in bulk ferroelectric ceramics via mesoscopic chemical inhomogeneity. *Adv. Sci.* 9, 2200998 (2022).
- [11] Kim, K., Lim, S.Y., Lee, J.U. et al. Suppression of magnetic ordering in XXZ-type antiferromagnetic monolayer NiPS₃. *Nat. Commun.* 10, 345 (2019).
- [12] Stöhr, J. & Siegmann, H. C. *Magnetism. Solid-State Sciences.* Springer, Berlin, Heidelberg 11, 479-520 (2006).

Reviewers' Comments:

Reviewer #4:

Remarks to the Author:

I believe the author addressed all the questions in the response letter and the revised manuscript, I recommend the publication.

REVIEWERS' COMMENTS

Reviewer #4 (Remarks to the Author):

I believe the author addressed all the questions in the response letter and the revised manuscript, I recommend the publication.

Reply: We thank the reviewer for carefully reading our work and recommending for publication.